# CMOOD: Concept-based Multi-label OOD Detection

**Zhendong Liu**[*]
*Independent Researcher*
*Lzd233a@gmail.com*

**Yi Nian**[*]
*University of Southern California*
*yinian@usc.edu*

**Yuehan Qin**[*]
*University of Southern California*
*yuehanqi@usc.edu*

**Henry Peng Zou**
*University of Illinois Chicago*
*pzou3@uic.edu*

**Li Li**
*University of Southern California*
*li.li02@usc.edu*

**Xiyang Hu**
*Arizona State University*
*xiyanghu@asu.edu*

**Reviewed on OpenReview:** *https://openreview.net/forum?id=EmoFJ8tcko*

## Abstract

How can models effectively detect out-of-distribution (OOD) samples in complex, multi-label settings without extensive retraining? Existing OOD detection methods struggle to capture the intricate semantic relationships and label co-occurrences inherent in multi-label settings, often requiring large amounts of training data and failing to generalize to unseen label combinations. While large language models have revolutionized zero-shot OOD detection, they primarily focus on single-label scenarios, leaving a critical gap in handling real-world tasks where samples can be associated with multiple interdependent labels. To address these challenges, we introduce CMOOD, a novel zero-shot multi-label OOD detection framework. CMOOD leverages pre-trained vision-language models, enhancing them with a concept-based label expansion strategy and a new scoring function. By enriching the semantic space with both positive and negative concepts for each label, our approach models complex label dependencies, precisely differentiating OOD samples without the need for additional training. Extensive experiments demonstrate that our method significantly outperforms existing approaches, achieving approximately 95% average AUROC on both VOC and COCO datasets, while maintaining robust performance across varying numbers of labels and different types of OOD samples. We release our code at https://github.com/boosLiu/COOD.

## 1 Introduction

As machine learning models become essential in a range of real-world applications, out-of-distribution (OOD) detection has gained increasing importance Yang et al. (2024). OOD detection is particularly critical in fields such as autonomous driving Elhafsi et al. (2023), medical diagnostics Huang et al. (2024), and surveillance Sultani et al. (2018), where detecting data that deviates from the training distribution is crucial to prevent safety risks or incorrect decisions Hendrycks et al. (2019b). Thus, developing robust OOD detection techniques is key to ensuring model reliability in unpredictable environments.

---

[*]Co-first authors

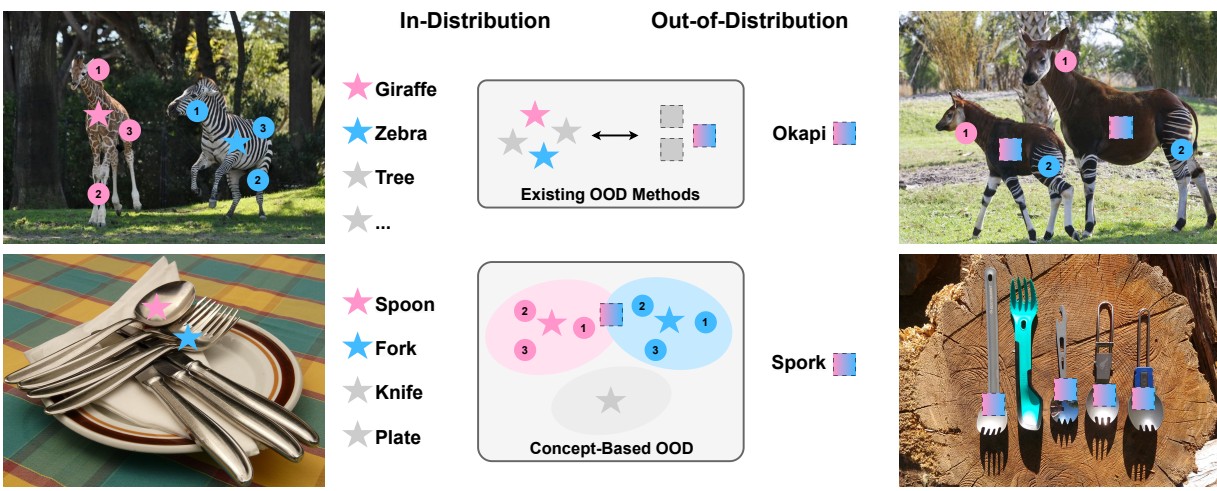

Figure 1: Motivation for CMOOD. Traditional methods struggle with complex multi-label cases. Our approach expands the label space with positive and negative concepts, enabling robust detection of complex OOD samples like "Okapi" and "Spork".

**Large Models for OOD Detection (LM-OOD)**. The rise of large models, particularly Vision-Language Models (VLMs) and MultiModal Large Language Models (MLLMs), has redefined the landscape of OOD detection Xu & Ding (2024). Traditional OOD detectors, such as Maximum Softmax Probability (MSP) and Mahalanobis distance-based techniques Lee et al. (2018), typically require extensive task-specific training on in-distribution (ID) data. In contrast, LLM-based methods leverage pre-trained knowledge, allowing for *zero-shot* and *few-shot* OOD detection. Models like CLIP Radford et al. (2021) exemplify this shift by achieving effective OOD detection with minimal task-specific training, relying on robust, pre-trained representations from multimodal datasets. Recent advancements, such as NegLabel Jiang et al. (2024), refine zero-shot OOD detection by introducing negative mining strategies to identify semantically meaningful OOD categories. This highlights that carefully selected negative labels can significantly improve detection performance without retraining. LM-OOD methods offer distinct advantages over traditional approaches, including strong performance in limited data, high adaptability across various tasks, and improved computational efficiency through reduced dependence on task-specific data preparation and retraining Ming et al. (2022); Miyai et al. (2024); Xu & Ding (2024).

**Limitations in Current LM-OOD**. Large models for OOD detection are primarily designed for single-label tasks, where each input corresponds to a single, definitive label Fort et al. (2021); Ming et al. (2022). However, many real-world applications are inherently multi-label in nature: medical imaging often requires identifying multiple coexisting conditions Kermany et al. (2018); Huang et al. (2024). Current state-of-the-art methods like NegLabel Jiang et al. (2024) encounter limitations in these multi-label contexts, as seen in Fig. 1: First, the complexity of semantic similarity computations increases significantly with multiple concurrent labels compared to single labels. Second, these methods assume OOD samples are semantically distant from ID classes, which may not hold in multi-label settings where novel combinations of known concepts could form valid OOD cases Wang et al. (2023); Nie et al. (2024). Additionally, these approaches struggle to effectively model complex label co-occurrence patterns and conditional dependencies, both crucial in multi-label contexts Zhang & Zhou (2014); Yu et al. (2014). These limitations highlight an essential gap in adapting LM-OOD methods for multi-label tasks, where it is necessary to differentiate between known and unknown label combinations within complex label spaces for practical application Miyai et al. (2024); Yang et al. (2024).

**Our Proposal: Extending LM-OOD to Multi-Label Settings**. In this work, we introduce CMOOD (concept-based OOD detection), the first multi-label OOD detection framework that leverages VLMs, specifically CLIP Wang et al. (2023), in a zero-shot setting. CMOOD addresses the challenges of multi-label OOD

detection by incorporating a concept-based label expansion strategy. It enriches the base label set with two fine-grained concepts: positive concepts, which capture complex semantic details related to ID classes, and negative concepts, which, filtered by a similarity threshold, introduce semantically distant features to strengthen ID-OOD separation. CMOOD embeds these expanded concept labels alongside the original base labels into a novel scoring function that accounts for both scenarios: OOD multi-label inputs that share more similarity with ID classes (e.g., Giraffe and Okapi) and those with less similarity to any ID class. This design enables precise detection of subtle distinctions between ID and OOD samples without requiring additional training. We define negative evidence relative to the compositional label space, allowing positive and negative semantics to co-exist within an image and reducing ambiguity when multi-label samples partially overlap with ID classes. Our technical contributions are summarized as follows:

- **Novel Multi-label OOD Detection**: We present a novel multi-label OOD detection framework based on the CLIP to achieve zero-shot detection without additional training.

- **Concept-based Label Expansion**: We introduce a concept-based label expansion that leverages positive and negative concepts for precise discrimination of OOD samples and provide interpretability in multi-label tasks.

- **Superior Performance and Efficiency**: Our method improves OOD accuracy while remaining lightweight; throughput measurements on CLIP-B/16 are reported in Appendix A.2.4.

## 2 Proposed CMOOD Method

### 2.1 Preliminaries on OOD Detection

In multi-label OOD detection, we determine whether an input image $I \in \mathcal{I}$ is ID or OOD. Let $\mathcal{B} = \{\ell_1, \ell_2, \ldots, \ell_{|\mathcal{B}|}\}$ denote the set of known ID classes, where each image $I$ can have multiple labels from $\mathcal{B}$. We define an ID scoring function $S_{\mathrm{ID}} : \mathcal{I} \to \mathbb{R}$ with the decision following criteria

$$S_{\mathrm{ID}}(I) \begin{cases} > \gamma, & \text{if } I \sim \mathcal{D}_{\mathrm{ID}}, \\ \leq \gamma, & \text{if } I \sim \mathcal{D}_{\mathrm{OOD}}, \end{cases} \tag{1}$$

where $\gamma$ is a threshold determined through validation, and $\mathcal{D}_{\mathrm{ID}}$, $\mathcal{D}_{\mathrm{OOD}}$ are the ID and OOD distributions, respectively. Throughout the paper we define OOD with respect to the semantic label space rather than the pretraining coverage of any model: inputs are ID if their semantics lie inside the predefined label set $\mathcal{B}$ and OOD otherwise. This formulation is model-agnostic, applies equally to CNN, ViT, and CLIP backbones, and motivates our concept expansion as a way to densify the label manifold where energy- or uncertainty-based scores alone under-specify the boundary in multi-label settings.

#### 2.1.1 Revisit VLM-based OOD Detection

VLMs bring powerful multimodal capabilities for OOD detection, allowing more flexible, context-aware identification of OOD samples Miyai et al. (2024); Xu & Ding (2024). Large-scale pre-trained VLMs, such as CLIP Radford et al. (2021) and GPT-4V Zhang et al. (2023), enable models to process images and text prompts together, enhancing adaptability and precision across diverse visual and textual domains Fort et al. (2021).

VLM-based OOD detection generally follows two primary approaches Xu & Ding (2024): *Prompting-based Detection*, which directly prompts VLMs to respond with OOD indicators Cao et al. (2023), and *Contrasting-based Detection*, which uses multimodal VLMs pre-trained with contrastive objectives to distinguish OOD samples Ming et al. (2022). This work focuses on the contrasting-based approach in multimodal contexts, as it is well-suited for enhancing OOD detection by amplifying distinctions between ID and OOD classes. Methods like `NegLabel` Jiang et al. (2024) and `NegPrompt` Li et al. (2024a) fall under this category. `NegLabel` enhances OOD detection by introducing negative labels to contrast with ID classes, while `NegPrompt` uses learned negative prompts to emphasize OOD differences by contrasting them with ID prompts. Our framework

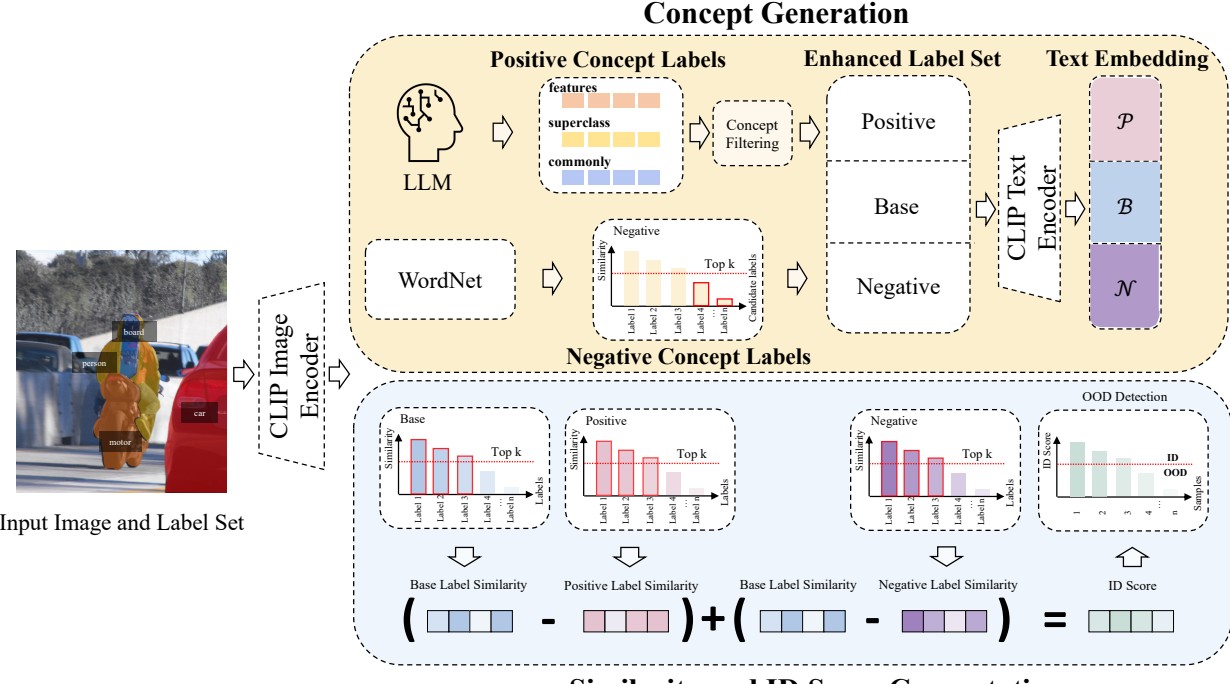

Figure 2: Overview of CMOOD. The Concept Generation module uses LLMs to expand base labels into positive ($\mathcal{P}$) and negative ($\mathcal{N}$) concept sets, enhancing the ID-OOD boundary. Positive concepts capture fine-grained, ID-aligned features, while negative concepts provide contrasting OOD-aligned features. The Similarity and ID Score Computation module encodes an input image and computes similarity scores. An ID score based on top-$k$ similarities then classifies the image for ID/OOD.

leverages the shared text–image alignment present across CLIP-style encoders (ResNet- and ViT-based), so it is not tied to a single implementation; the same scoring applies across these backbones without architectural changes.

**Formal Definition**. The OOD detection score $S(x)$ for a sample $x$ is defined as:

$$S_{OOD}(x) = \frac{\sum_{i \in Y} \exp\big(\text{sim}(h, e_i)\big)}{\sum_{i \in Y} \exp\big(\text{sim}(h, e_i)\big) + \sum_{j \in Y^-} \exp\big(\text{sim}(h, e_j^-)\big)}, \qquad (2)$$

where $h$ is the embedding of $x$; $e_i$ and $e_j^-$ are embeddings for ID and negative representations (`NegPrompt` Li et al. (2024a), `NegLabel` Jiang et al. (2024)) from sets $Y$ and $Y^-$, respectively; $\text{sim}(h, e)$ denotes the similarity (e.g., cosine similarity) between embeddings. The numerator measures similarity to ID representations, while the denominator amplifies the detection signal with both ID and negative similarities. Note that our definition is a little different: a smaller $S(x)$ means a larger probability of $x$ being an OOD sample.

**Limitations in Current Approaches**. Current methods like NegLabel and NegPrompt assume that OOD classes are semantically distant from ID classes, which may not hold in real-world scenarios. OOD samples can closely resemble ID classes, particularly in fine-grained distinctions (e.g., similar dog breeds), limiting the effectiveness of a purely semantic distance-based approach.

Moreover, extending single-label OOD detection methods Sun et al. (2021); Liu et al. (2020) to multi-label contexts presents additional challenges. Multi-label settings often involve significant semantic overlap between labels and co-occurring ID and OOD labels, complicating the extraction of effective negative labels

or prompts. These issues highlight the need for advanced detection techniques to capture the complex dependencies and relationships inherent in multi-label OOD detection tasks.

## 2.2 Overview of CMOOD Method

In fact, it is not new to use LLM to assist in generating descriptions and then using CLIP models for classification tasks like DCLIP Menon & Vondrick (2022). However, there are two problems with the application of this method: (1) the generated description only contains texts with similar semantic meanings, and the description of the text space is not comprehensive. (2) When considering the classification task, the OOD detection task does not bring improvement, and even brings negative benefits as show in Table 1.

The CMOOD framework improves OOD detection by refining the decision boundary between ID and OOD samples using fine-grained concepts. By introducing positive and negative concept sets, $\mathcal{P}$ and $\mathcal{N}$, CMOOD leverages LLMs to add contextual information around ID samples, thereby enhancing sensitivity to OOD cases.

CMOOD is built on two core components (see Fig. 2):

1. **Concept Generation and Similarity Measure**: To distinguish ID from OOD samples, we generate positive concepts ($\mathcal{P}$) and negative concepts ($\mathcal{N}$) that are closely and distantly related to ID classes, respectively. Positive concepts, mined using LLMs, capture domain-relevant features of ID samples at a fine-grained level, while negative concepts are chosen based on their semantic distance from ID classes to enhance contrast.

2. **Similarity and ID Score Computation**: For each input image $I$, we compute similarity scores with these concept sets and the base label set $\mathcal{B}$, forming a robust semantic space for evaluating ID-OOD relationships. After this, we define an ID score $S_{\text{ID}}(I)$ that aggregates the image's alignment with base concepts ($\mathcal{B}$) and contrasts it with the positive and negative concepts ($\mathcal{P}$ and $\mathcal{N}$). When this score falls below a predefined threshold $\gamma$, the image is classified as OOD. This scoring method sharpens the decision boundary by leveraging contrasts with positive and negative concepts giving more accurate OOD detection.

Concept probes approximate the boundary of the semantic support induced by $\mathcal{B}$: attribute anomalies or label noise that remain within this support are treated as ID, whereas semantically distant positives/negatives refine the boundary for true distribution shifts without redefining the label space.

The pseudocode in Algo. 1 outlines the full procedure for CMOOD. To formalize the OOD detection decision, we compute an ID score and classify an image $X$ as OOD if this score falls below a threshold $\gamma$. Specifically, the decision function $\tilde{Y}$ is defined as follows:

$$
\begin{aligned}
\tilde{Y} &= \mathbb{I}(S_{\text{ID}}(h, P, N) < \gamma), \quad \text{where} \quad h = f_{\text{img}}(X), \\
P &= f_{\text{text}}(\text{prompt}(\mathcal{P})), \quad N = f_{\text{text}}(\text{prompt}(\mathcal{N})).
\end{aligned}
\tag{3}
$$

Here, $h$ is the image embedding for $X$, obtained using an image encoder $f_{\text{img}}$, while $P$ and $N$ are the positive and negative concept embeddings generated via text prompts for concept sets $\mathcal{P}$ and $\mathcal{N}$, respectively. The indicator function $\mathbb{I}$ produces the final OOD classification.

**Advantages.** The proposed CMOOD approach is computationally efficient, leveraging pre-trained embeddings without requiring additional model training. By incorporating positive and negative concepts, the model gains enhanced semantic understanding, allowing for clearer differentiation between ID and OOD samples. The focus on top-$k$ similarity values makes the method robust to noise, ensuring stable performance across varied datasets and label sets.

## 2.3 Concept Generation

**Motivation**. The concept generation process in CMOOD creates a rich semantic space that strengthens ID-OOD distinctions. By constructing a set of positive concepts, $\mathcal{P}$, aligned with ID samples, and a set of

---

**Algorithm 1** Multi-Label OOD Detection with CMOOD

---

**Require:** Image $I$, base label set $\mathcal{B}$, threshold $\gamma$, top-$k$ parameter $k$
**Ensure:** Classification of $I$ as ID or OOD
  1: **Concept Generation** (§2.3): Generate positive concepts $\mathcal{P}$ and negative concepts $\mathcal{N}$ using an LLM.
  2: **Embedding Computation**: Compute embeddings for $I$, as well as for each label in $\mathcal{B}$, $\mathcal{P}$, and $\mathcal{N}$.
  3: **Top-$k$ Similarity Calculation** (§2.4): Calculate the top-$k$ mean similarity scores, $\mu_k(\mathcal{B}, I)$, $\mu_k(\mathcal{P}, I)$, and $\mu_k(\mathcal{N}, I)$.
  4: **ID Score Computation** (§2.4): Using the top-$k$ similarities, compute the ID score $S_{\mathrm{ID}}(I)$ according to Eq. equation 7.
  5: **Decision**: **if** $S_{\mathrm{ID}}(I) > \gamma$ **then** classify $I$ as ID; **else** classify $I$ as OOD.

---

negative concepts, $\mathcal{N}$, that enhance the contrast with OOD samples, we can refine the decision boundary between ID and OOD. The sets $\mathcal{P}$ and $\mathcal{N}$ are complementary: $\mathcal{P}$ effectively captures multi-label inputs that share similarities with ID classes, while $\mathcal{N}$ captures multi-label inputs whose components are all dissimilar from ID samples. Together, these complementary concept sets enable CMOOD to model both subtle distributional shifts and drastic deviations, leading to improved OOD detection across varying degrees of distribution shift.

**Positive Concept Mining $\mathcal{P}$.** To build a comprehensive set of positive concepts that accurately represents ID characteristics, we use a two-stage approach inspired by LF-CBM Oikarinen et al. (2023). This process enriches each base label by capturing specific attributes that reinforce its identity within the ID class.

1. **Concept Querying**: We prompt GPT-4 to generate concepts in three distinct contexts for each target object, using structured prompts tailored to elicit three categories: features ($\mathrm{prompt}_F$), superclasses ($\mathrm{prompt}_S$), and commonly associated items ($\mathrm{prompt}_C$). This contextual querying ensures that each concept captures detailed, complex information, improving the clarity and consistency of the concept pool. Each prompt is crafted to encourage concise, relevant responses without qualifiers, which enhances precision, detailed in Appendix §A.1.1.

2. **Concept Filtering**: After generating these candidate concepts, we apply filtering criteria to retain only the most distinctive and relevant features. This step is essential to ensure that the concept set effectively captures the ID characteristics required to differentiate ID from OOD. Our filtering process, detailed in the Appendix §A.1.1, refines the generated concepts into a cohesive and meaningful set of positive labels.

The final positive concept set, $\mathcal{P}$, is constructed as:

$$\mathcal{P} = \bigcup_{c_i \in \mathcal{B}} \left( \mathrm{prompt}_F(c_i) \cup \mathrm{prompt}_S(c_i) \cup \mathrm{prompt}_C(c_i) \right), \tag{4}$$

where $\mathcal{B}$ denotes the set of ID labels, and each $c_i \in \mathcal{B}$ is queried to generate its positive concept labels.

**Negative Concept Mining $\mathcal{N}$.** To reinforce the distinction between ID and OOD samples, we develop a set of negative concepts, $\mathcal{N}$, using a process called NegMining Jiang et al. (2024). This approach helps us identify concepts that are semantically distant from ID labels, forming a contrasting boundary that enhances OOD detection.

Starting with a large collection of words from a lexical database like WordNet, we create a candidate label space $\mathcal{N}^c = \{n_1, n_2, \ldots, n_C\}$. For each candidate label $\tilde{n}_i$ in this set, we calculate a similarity score based on its cosine similarity with the entire ID label set, ensuring that the chosen negative concepts have minimal alignment with ID labels.

- For each ID label $l \in \mathcal{B}$, we obtain its text embedding $e_l$ using a text encoder $f^{\mathrm{text}}$.

- For each candidate $\tilde{n}_i \in \mathcal{N}^c$, we compute its embedding:

$$\tilde{e}_i = f^{\mathrm{text}}(\mathrm{prompt}(\tilde{n}_i)). \tag{5}$$

**Figure 3:** t-SNE figure of label and concept text embeddings, along with corresponding image examples. In scenarios with multiple labels and objects, it is difficult to model the OOD detection problem using a single similarity measure. Instead, the COOD method is employed to address this issue.

We select the top candidates with the smallest similarity scores relative to the ID label set, forming a set of negative labels that are maximally distant from ID concepts.

The final negative concept set is defined as:

$$\mathcal{N} = \{n \mid \text{Sim}(\tilde{n}_i, \mathcal{B}) < \tau_i\},$$
$$\tau_i = \text{percentile}_\eta\left(\{\text{Sim}(\tilde{e}_i, e_l)\}_{l \in \mathcal{B}}\right), \tag{6}$$

where $\text{Sim}(\tilde{n}_i, \mathcal{B})$ represents the similarity between candidate $\tilde{n}_i$ and the ID set $\mathcal{B}$, and $\tau_i$ is the $\eta$-th percentile of similarity scores, providing robustness against outliers.

By expanding the base label set $\mathcal{B}$ to include positive concepts $\mathcal{P}$ and contrasting them with negative concepts $\mathcal{N}$, we create an enriched semantic space. This structure allows for more precise discrimination between ID and OOD samples, capturing both the core characteristics of the ID classes and intra-class and inter-class similarity effectively.

## 2.4 Similarity and ID Score Computation

After generating our positive and negative concepts, CMOOD computes an ID score for each input image $I$, based on its similarity with the positive concept set $\mathcal{P}$, negative concept set $\mathcal{N}$, and base labels $\mathcal{B}$. This score helps determine if the image exhibits more ID or OOD characteristics. As shown in Figure 3, the positive concepts are distributed around the base labels, while the negative concepts are located at a relatively distant position in the text embedding space, showing a significant distinction. Therefore, to capture this distinction, we model the problem by adjusting the base label-image similarity through two components: (1) Subtracting the intra-class similarity between nearby positive concepts and the image, which captures within-distribution variations. Positive concepts are useful when identifying multi-label OOD inputs where it has some similarities with ID class like Giraffe and Okapi in Fig. 1. (2) Subtracting the inter-class similarity between distant negative concepts and the image, which captures cross-distribution relationships. This is to penalize the score when multi-label OOD input is not relevant to ID class.

**ID Score Definition.** The ID score $S_{\text{ID}}(I)$ for an image $I$ is:

$$S_{\text{ID}}(I) = \underbrace{[\mu_k(\mathcal{B}, I) - \mu_k(\mathcal{P}, I)]}_{S_{\text{A}}} + \underbrace{[\mu_k(\mathcal{B}, I) - \mu_k(\mathcal{N}, I)]}_{S_{\text{B}}} \quad (7)$$

where $\mathcal{B}$ are the base labels, $\mathcal{P}$ are the positive concepts, and $\mathcal{N}$ are the negative concepts.

In general, when an out-of-distribution (OOD) input has some similarity to an in-distribution (ID) class, we penalize $S_A$, whereas when an OOD input has no similarity to any ID class, we penalize $S_B$. However, for ID inputs, both $S_A$ and $S_B$ are larger. Consequently, ID inputs receive higher scores, while OOD inputs receive lower scores. We include theoretical justification in §2.5 and provide corresponding visualizations in Figure 5.

**Top-$k$ Mean Similarity Calculation.** The term $\mu_k(\mathcal{S}, I)$ in Eq. (7) represents the top-$k$ mean similarity score between the image $I$ and the label set $\mathcal{S}$, calculated as:

$$\mu_k(\mathcal{S}, I) = \frac{1}{k} \sum_{i=1}^{k} \text{sim}(h, e_i) \quad (8)$$

where $h$ is the embedding of image $I$, and $e_i$ are the embeddings of the top-$k$ closest elements in set $\mathcal{S}$. Here, $\mathcal{S}$ can be any of $\mathcal{B}$ (base labels), $\mathcal{P}$ (positive concepts), or $\mathcal{N}$ (negative concepts). The similarity function $\text{sim}(h, e_i)$ (e.g., cosine similarity) captures the alignment between the image embedding $h$ and each of the concept embeddings $e_i$.

This top-$k$ similarity measure emphasizes the most relevant matches between the image and the concepts in each set, minimizing the influence of less relevant or noisy features and enhancing robustness in scoring.

**Interpretation of the ID Score.** The scoring function $S_{\text{ID}}(I)$ captures the degree of alignment of an image within ID or with OOD characteristics. For ID samples, we anticipate high alignment with base labels $\mathcal{B}$, yielding a high $S_{\text{ID}}(I)$. Conversely, OOD samples will likely align more closely with negative concepts $\mathcal{N}$, resulting in a lower ID score. By focusing on the top-$k$ similarities, this approach enhances robustness to noise and emphasizes the most semantically relevant features.

This mechanism refines the boundary between ID and OOD samples by leveraging positive and negative contrasts to base labels, resulting in more accurate and reliable OOD detection.

## 2.5 Theoretical Justification

**Lemma 1** (Separability of ID-OOD). *The scoring function can distinguish between in-distribution (ID) and out-of-distribution (OOD) samples based on their semantic relationships by appropriately selecting weights $w_P$ and $w_{\mathcal{N}}$.*

*Proof.* We assume the following bounds on the similarity scores $\mu_k(\cdot, \cdot)$ between an image $I$ and class representations in different contexts when $I$ is either ID or OOD:

$$0 \leq \mu_k(\mathcal{B}, I_{\text{OOD}}) \leq a < \mu_k(\mathcal{B}, I_{\text{ID}}) \leq A \leq 1,$$
$$0 \leq \mu_k(\mathcal{P}, I_{\text{OOD}}) \leq b < \mu_k(\mathcal{P}, I_{\text{ID}}) \leq B \leq 1,$$
$$0 \leq \mu_k(\mathcal{N}, I_{\text{ID}}) \leq c < \mu_k(\mathcal{N}, I_{\text{OOD}}) \leq C \leq 1.$$

To establish separability between ID and OOD samples, we examine bounds on the scoring function for in-distribution and out-of-distribution images.

1. **Lower Bound for $S_{\text{ID}}$:** Based on the assumed bounds, the lower bound of the scoring function $S_{\text{ID}}$ for ID samples is: $S_{\text{ID}}^{\text{lower}} = w_B a - w_P B - w_{\mathcal{N}} c$.

2. **Upper Bound for $S_{\text{ID}}$:** The upper bound of the scoring function $S_{\text{ID}}$ for OOD samples is: $S_{\text{ID}}^{\text{upper}} = w_B a - w_{\mathcal{N}} C$.

3. **Separability Condition:** For the scoring function to effectively distinguish between ID and OOD samples, we require: $S_{\text{ID}}^{\text{lower}} - S_{\text{ID}}^{\text{upper}} > 0$. Substituting the expressions for $S_{\text{ID}}^{\text{lower}}$ and $S_{\text{ID}}^{\text{upper}}$, this simplifies to: $w_P B < w_{\mathcal{N}}(C - c)$.

By selecting $w_P$ and $w_{\mathcal{N}}$ to satisfy this condition, we ensure that the scoring function $S_{\text{ID}}$ can differentiate between ID and OOD samples based on semantic similarity. □

**Implications.** This lemma establishes a theoretical basis for using the scoring function to distinguish ID from OOD samples by focusing on top-$k$ similarities, enhancing robustness by minimizing the effect of weakly relevant labels.

Table 1: OOD Detection Results on Various Datasets. Our ViT-based CLIP model achieves strong performance with the lowest FPR95 and highest AUROC across most datasets, outperforming standard OOD methods. The ResNet-based CLIP variant also performs competitively.

| ID Dataset | Pascal VOC | | | | COCO | | | | ImageNet | |
|---|---|---|---|---|---|---|---|---|---|---|
| OOD Dataset | ImageNet22k | | Textures | | ImageNet22k | | Textures | | Textures | |
| Method | FPR95↓ | AUROC↑ | FPR95↓ | AUROC↑ | FPR95↓ | AUROC↑ | FPR95↓ | AUROC↑ | FPR95↓ | AUROC↑ |
| **ResNet-based Multi-label Classifier** | | | | | | | | | | |
| MaxLogit* Hendrycks et al. (2019a) | 36.32 | 91.04 | 12.36 | 96.22 | 44.47 | 87.13 | 19.83 | 95.31 | 57.09 | 86.71 |
| MSP* Hendrycks & Gimpel (2016) | 69.85 | 78.24 | 41.81 | 89.76 | 82.15 | 67.47 | 65.21 | 81.88 | 68.00 | 79.61 |
| ODIN* Liang et al. (2017) | 36.32 | 91.04 | 12.36 | 96.22 | 54.51 | 84.92 | 33.15 | 90.71 | 50.23 | 85.62 |
| (Joint)-Energy* Liu et al. (2020); Wang et al. (2021) | 31.96 | 92.32 | 10.87 | 96.78 | 41.81 | 90.30 | 17.72 | 96.07 | 53.72 | 85.99 |
| **Ours (ResNet-based CLIP)** | 25.28 | 93.32 | **8.76** | **97.79** | 26.83 | 93.14 | **10.53** | **97.17** | 43.72 | 91.68 |
| **ViT-based CLIP** | | | | | | | | | | |
| MSP† Hendrycks & Gimpel (2016) | 86.35 | 75.42 | 64.74 | 85.46 | 59.27 | 87.30 | 45.57 | 90.35 | 64.96 | 78.16 |
| (Joint)-Energy† Liu et al. (2020); Wang et al. (2021) | 81.93 | 76.59 | 90.18 | 78.23 | 65.39 | 85.20 | 76.01 | 82.40 | 51.18 | 88.09 |
| DCLIP*Menon & Vondrick (2022) | 96.96 | 61.59 | 97.94 | 65.26 | 89.50 | 71.43 | 72.77 | 83.34 | 91.08 | 70.75 |
| MCM†* Ming et al. (2022) | 73.81 | 80.37 | 53.67 | 88.52 | 63.34 | 86.10 | 49.22 | 89.11 | 57.77 | 86.11 |
| GL-MCM†* Miyai et al. (2024) | 72.98 | 81.76 | 64.74 | 86.96 | 48.96 | 88.50 | 45.06 | 89.70 | 57.41 | 83.73 |
| NegLabel*Jiang et al. (2024) | 35.83 | 91.18 | 43.14 | 89.72 | 33.24 | 90.19 | 47.33 | 85.10 | 43.56 | 90.22 |
| MCM+SeTAR†* Li et al. (2024b) | 48.25 | 92.08 | 40.44 | 93.58 | 73.55 | 80.43 | 47.33 | 89.58 | 55.83 | 86.58 |
| GL-MCM+SeTAR†* Miyai et al. (2024); Li et al. (2024b) | 31.47 | 94.31 | 20.35 | 96.36 | 65.30 | 81.38 | 42.05 | 89.81 | 54.17 | 84.59 |
| CLIPN*Wang et al. (2023) | 64.78 | 82.46 | 37.42 | 92.63 | 44.63 | 89.21 | 25.37 | 94.08 | 40.83 | 90.93 |
| CLIPScope*Fu et al. (2024) | 70.86 | 85.51 | 77.42 | 83.04 | 55.90 | 87.63 | 73.63 | 78.71 | **38.37** | **91.41** |
| **Ours (ViT-based CLIP)** | **23.87** | **94.32** | 21.58 | 95.14 | **20.37** | **95.07** | 21.63 | 94.53 | 39.41 | 91.10 |

# 3 Experimental Results

## 3.1 Experimental Setup

**Datasets and Settings.** We evaluate our proposed multi-label OOD detection method on widely used datasets. Following previous work Ming et al. (2022); Miyai et al. (2024); Hendrycks et al. (2019a), we adopt two widely used real-world datasets, ImageNet-1k Deng et al. (2009) and Pascal VOC Everingham et al. (2015) and MS-COCO Lin et al. (2014) as the InD datasets. These datasets provide a diverse set of object categories with high-quality annotations, making them suitable for evaluating multi-label OOD detection and complex OOD detection tasks. For OOD datasets, we preprocess iNaturalist, SUN, Places, and Texture following the methodology in Jiang et al. (2024), and use ImageNet22k data following Hendrycks et al. (2019a). These preprocessing steps ensure no class overlaps between InD and OOD datasets. For evaluation, we use two standard OOD detection metrics: **AUROC** and **FPR@95**. See Appendix A.2.1 for more

Table 2: Datasets used in the experiments for OOD detection.

| | Dataset Name | Number of Samples |
|---|---|---|
| InD | ImageNet-1k Validation | 50,000 |
| | Pascal VOC Test | 16,135 |
| | COCO2017 Test | 40,670 |
| OOD | ImageNet22K | 18,335 |
| | Textures | 5,640 |
| | iNaturalist | 10,000 |
| | SUN | 10,000 |
| | Places | 10,000 |

experimental setup details.

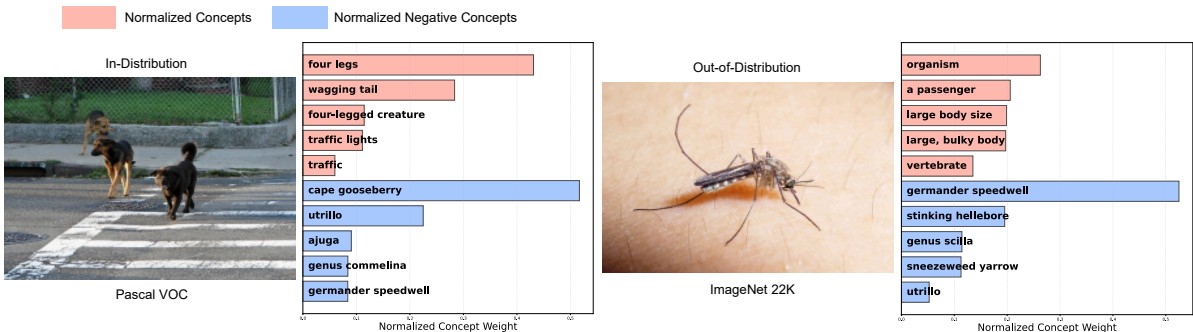

Figure 4: Analysis of CMOOD on ID (left) and OOD (right) examples. For ID sample (dogs), positive concepts (e.g.,"four-legged creature") receive high weights, confirming ID alignment. For OOD samples (mosquito), negative concepts dominate.

**Implementation Details.** Our method is implemented in PyTorch and evaluated on an NVIDIA L20 GPU; with batch size 8 on CLIP-B/16 we achieve 203 images/second (4.93 ms/img) and 1565 MB memory, improving wall-clock throughput by 28–44% over baselines that use 1807 MB. As a zero-shot approach, our method leverages pre-computed concept embeddings and does not require any additional training. We fix the Top-$k$ parameter within the stable range $[0.05, 0.2]$ observed in Table 4 and do not tune it on validation data, keeping the method training-free. Efficiency breakdowns and FLOPs are reported in Appendix A.2.4. This efficient design allows for parallelized OOD detection alongside zero-shot classification tasks, making it suitable for deployment in real-time applications. For reproducibility, all hyperparameters and settings follow standard configurations, and further implementation details are provided in Appendix A.2.1.

## 3.2 Experimental Results and Analysis

**OOD Detection Performance Comparison.** Table 1 presents a comparison of the OOD detection performance of CMOOD against several baseline approaches. Where **\*** represents our reproduction of the missing results from the original paper, and **†** means that we used the values from the original paper, and **†\*** represents the best value we take from the replicated results and the reported results in the original paper. The gray boxes represent our approach. In addition, **(Joint)-Energy** denotes the better of the JointEnergy (Wang et al., 2021) method and the Energy (Liu et al., 2020) method. **Bold** values indicate the best results. *Our method demonstrates strong OOD detection performance across both ResNet-based and ViT-based architectures.* Specifically, our ResNet-based CLIP achieves superior results, particularly on the Pascal VOC ID dataset and Textures OOD dataset, with FPR95 of 8.76% and AUROC of 97.79%, outperforming all baseline methods. These suggest that the ResNet-based architecture may be particularly effective in handling texture-oriented datasets.

On the ViT-based CLIP, our method achieves the lowest FPR95 of 23.87% on the Pascal VOC dataset and outperforms competing methods on the COCO dataset, achieving an impressive AUROC of 95.07%. The ViT-based approach performs exceptionally well across most datasets, indicating that the ViT architecture enhances OOD detection across diverse image distributions, while still maintaining strong robustness on texture-based datasets. For more OOD dataset like NINCO Bitterwolf et al. (2023) and SSB Vaze et al. (2022), please see Table 3. Our CMOOD also demonstrates strong competitiveness.

Partial OOD setting. To test mixed images containing both ID and unseen objects, we resplit Pascal VOC into 11 ID classes (e.g., aeroplane, bicycle, dog) and 9 OOD classes (e.g., chair, sofa, train) without retraining. On 500 validation images, the ID score shows graceful degradation: pure ID samples score 3.15, mixed ID/OOD samples score 1.92, and pure OOD samples score $-0.20$, confirming image-level scoring aggregates signals across co-occurring objects (Appendix A.2.6, Table 15).

Table 3: AUROC / FPR@95 for more OOD datasets.

| ID Dataset | Method | NINCO | SSB-easy | SSB-hard |
|---|---|---|---|---|
| COCO2017 | CMOOD | 85.89 / 40.58 | 82.71 / 54.30 | 87.72 / 45.19 |
| Pascal VOC | CMOOD | 88.79 / 42.17 | 88.75 / 43.71 | 72.70 / 59.62 |
| ImageNet | MSP | 69.32 / 81.09 | 80.06 / 80.79 | 55.83 / 93.27 |
| | MCM | 68.80 / 83.39 | 80.20 / 78.15 | 55.83 / 93.27 |
| | NegLabel | 72.96 / 72.81 | 71.22 / 80.13 | 55.40 / 88.94 |
| | CMOOD | **77.40 / 69.91** | **84.04 / 69.54** | **64.67 / 81.25** |

Importantly, our approach operates as a *zero-shot* method, requiring no additional training or complex parameter tuning. This makes it highly efficient with low computational overhead. This contrasts with other multi-label classifiers requires training. For method without training, such as SeTAR Li et al. (2024b), which requires extensive parameter search time, impacting practical deployment. Overall, our method provides an efficient and effective zero-shot solution for OOD detection, yielding state-of-the-art performance with minimal overhead on both architectures.

### 3.3 Visual Explainability of CMOOD Components

Fig. 3 and 4 present a detailed analysis of CMOOD through t-SNE visualization Van der Maaten & Hinton (2008) of text embeddings and qualitative examples. CMOOD's explainability is a key feature, allowing us to understand how specific concepts contribute to OOD detection.

**t-SNE Visualization.** The t-SNE plot on the Fig. 3 visualizes the embeddings of concepts from our model, showing a clear separation between ID labels (Pascal VOC, shown in red) and OOD concepts. Notably, the embeddings for positive concepts $\mathcal{P}$ (shown in red) form distinct clusters separate from negative concepts $\mathcal{N}$ (shown in blue), while the black points represent the base labels $\mathcal{B}$. This clustering reflects the model's ability to encode meaningful semantic information that distinguishes ID from OOD samples. The clustering of positive concepts around ID points and the scattering of negative concepts in more distant regions confirm that the model effectively captures relationships that contribute to its decision-making process.

**Qualitative Examples.** Fig. 4 shows example images with corresponding bar plots, illustrating the concept weights associated with each image. In each bar plot, positive concepts are marked in red, while negative concepts are marked in blue. This visual representation offers insight into which semantic features are influencing the OOD score, making it easier to interpret why certain samples are classified as ID or OOD.

### 3.4 Ablation Studies and Additional Analysis

To assess the contributions of different components in our approach, we conduct an ablation study with variations in the parameters $S_A$ and $S_B$ on the Pascal VOC and COCO datasets, as shown in Table 4. For more experiments on different model architectures and OOD score functions, please refer to Appendix A.2. Recall that we defined the score function in Eq. (7). Enabling both $S_A$ and $S_B$ yields the best performance across metrics. For example, on Pascal VOC, activating both scores reduces FPR95 from 36.12% to 24.78% and boosts AUROC from 91.72% to 94.27%. This improvement indicates that $S_A$ and $S_B$ capture complementary aspects of the data, thereby enhancing the model's ability to differentiate between ID and OOD samples effectively. Across the same study, varying Top-$k$ within $[0.05, 0.2]$ shifts AUROC by at most 3% on Pascal VOC and 2.7% on COCO, so we keep it fixed rather than tuned per dataset (Appendix A.2.5).

Table 4: Ablations on Pascal VOC and COCO datasets. Using both $S_A$ and $S_B$ together yields the best results.

| $S_A$ | $S_B$ | Pascal VOC ID | | COCO ID | |
|---|---|---|---|---|---|
| | | FPR95 | AUROC | FPR95 | AUROC |
| ✗ | ✓ | 36.12 | 91.72 | 48.87 | 89.49 |
| ✓ | ✗ | 36.43 | 91.73 | 33.70 | 90.09 |
| ✓ | ✓ | 24.78 | 94.27 | 29.08 | 92.39 |

## 4 Related Work

### 4.1 Zero-Shot and Multi-Label OOD Detection

With the rise of contrastive learning-based cross-modal models like CLIP, traditional OOD detection methods face new challenges, particularly in zero-shot settings. In addition, the increasing complexity of image understanding tasks has introduced challenges in multi-label OOD detection.

For OOD detection in CLIP-based settings, several methods have been proposed, including both zero-shot and fine-tuning approaches. For example, ZOC (Esmaeilpour et al., 2022) introduces a trainable generator and additional data to create extra OOD labels. Maximum Concept Matching (MCM) (Ming et al., 2022) uses textual embeddings of ID classes as concept prototypes, calculating an MCM score based on the cosine similarity between image and textual features to identify OOD instances. NegLabel (Jiang et al., 2024) enhances OOD detection by incorporating a large set of negative labels, which are semantically distant from in-distribution (ID) labels, thereby improving the model's ability to distinguish OOD samples. GL-MCM (Miyai et al., 2023) builds upon CLIP by considering both global and local visual-text alignments in the detection process. SeTAR (Li et al., 2024b) applies selective low-rank approximation to weight matrices with a greedy search algorithm for optimized OOD detection. NegPrompt (Nie et al., 2024) introduces transferable negative prompts for OOD detection, where each ID class is paired with learnable negative prompts, improving OOD detection by associating OOD samples more closely with negative prompts. CLIPN Wang et al. (2023) and CLIPScope also similarly develop additional labels to enhance OOD detection performance. However, these approaches lack a discussion of multi-label scenarios.

In multi-label tasks, some researchers have explored OOD detection as well. The JointEnergy-based method (Wang et al., 2021) is one of the first to address OOD detection in multi-label scenarios. A later improvement (Mei et al., 2024) further refines this approach. Additionally, YolOOD (Zolfi et al., 2024) explores OOD detection in multi-label tasks, using an object detection model as the backbone. However, these methods do not explore CLIP-based architectures. Methods utilizing multi-label data with CLIP, such as GL-MCM (Miyai et al., 2023) and SeTAR (Li et al., 2024b), typically focus on curated datasets and fail to address the challenge of detecting OOD samples when multiple classes co-exist in the same image. Moreover, GL-MCM+SeTAR (Miyai et al., 2024; Li et al., 2024b) requires fine-tuning. In contrast, our work is the first to comprehensively explore how to effectively perform zero-shot OOD detection in multi-label tasks using CLIP models.

Table 5: Representative OOD Detection Performance on Pascal VOC (Textures) with Different Architectures and Distance Metrics

| Architecture | Distance Metric | FPR95 | AUROC |
|---|---|---|---|
| ResNet50 | mean | 8.76 | 97.79 |
| ResNet50x16 | mean | 18.54 | 95.89 |
| ViT-B/16 | mean | 21.58 | 95.14 |
| ViT-B/16 | percentile (0.75) | 26.62 | 93.99 |
| ViT-L/14 | mean | 24.28 | 94.55 |

### 4.2 Concept Bottlenecks

Concept bottlenecks have emerged as a promising approach to improve model interpretability and robustness by explicitly modeling the intermediate concepts that influence predictions. Koh et al. (2020) proposed using a feature extractor and a concept predictor to generate these "bottleneck" concepts, which are then used in a final predictor to determine class labels. Oikarinen et al. (2023) introduced the Label-Free Concept Bottleneck Model (LF-CBM), which generates concepts without requiring manually labeled concept data. Instead, it employs unsupervised methods to automatically identify and extract meaningful concepts from the data. Xu et al. (2024) introduced Energy-Based Concept Bottleneck Models (ECBM), which learn positive and negative concept embeddings to capture high-order nonlinear interactions between concepts, enabling richer concept explanations. Probabilistic Concept Bottleneck Models (ProbCBMs) (Kim et al., 2023) integrate uncertainty estimation with concept predictions. Recently, various other CBM variants have emerged, expanding the versatility of this approach. While these methods have been effective in extracting concepts and describing multiple labels within images, to the best of our knowledge, none have directly applied these concept-based techniques to multi-label and complex OOD detection tasks. Our approach focuses on the novel application of concept bottlenecks specifically for improving OOD detection in such settings. In addition, our approach addresses semantic ambiguity in overlapping labels via concept expansion and anchor.

## 5 Conclusion, Limitations, and Broader Impact

We introduced CMOOD, a novel framework that leverages concept-based reasoning for zero-shot Out-of-Distribution (OOD) detection in complex multi-label environments. Our approach sets a new state of the art, achieving over 95% average AUROC on VOC and COCO by effectively distinguishing In-Distribution data from challenging OOD datasets like ImageNet and Texture. Through comprehensive experiments, we also demonstrated the method's high degree of explainability and robustness. This work validates that dissecting scenes into fine-grained concepts is a powerful and efficient paradigm for multi-label OOD detection.

Our work builds upon the powerful capabilities of large-scale language models. Like all methods leveraging such foundation models, the performance of CMOOD is correlated with the quality of the embedding space and the richness of the provided concepts. This connection does not represent a limitation of our framework, but rather highlights a frontier for future innovation. We identify two promising research trajectories: (1) Learning concept vocabularies directly from data or external knowledge graphs, moving towards a more adaptive and scalable system beyond concepts from language models. (2) Adapting our method to specialized domains. In medical image processing, for instance, this could enable the detection of rare diseases as OOD.

CMOOD enhances AI reliability in safety-critical applications, such as healthcare and autonomous systems, by improving robustness against unknown data. This directly mitigates the risks of false detections, which can have severe consequences, thus contributing to safer performance in dynamic, high-stakes environments.

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

# A   Appendix

## A.1   Method Implementation Details

### A.1.1   Concept Generation

Concept generation plays a crucial role in our framework by leveraging semantic features to improve interpretability and OOD detection. For each label in the InD datasets, we utilize a set of prompts to generate associated concepts, follow the prompts in LF-CBM (Oikarinen et al., 2023) and CEIR (Cui et al., 2023). The prompts are carefully designed to capture three types of semantic information: **important features**, **superclasses**, and **common surroundings**, efficiently and precisely. The specific prompts used are as follows:

- **Important Features:**

  ```
  List the most important features
  for recognizing something
  as a "goldfish":
  - bright orange color
  - a small, round body
  - a long, flowing tail
  - a small mouth
  - orange fins

  List the most important features
  for recognizing something
  as a "{}":
  ```

- **Superclasses:**

  ```
  Give superclasses for
  the word "tench":
  - fish
  - vertebrate
  - animal

  Give superclasses for
  the word "{}":
  ```

- **Common Surroundings:**

  ```
  List the things most commonly
  seen around a "tench":
  - a pond
  - fish
  - a net
  - a rod
  - a reel
  - a hook
  - bait

  List the things most commonly
  seen around a "{}":
  ```

Using these prompts, we extract meaningful semantic features for each class, which are then processed and refined to ensure relevance and diversity. For example, from the COCO2017 dataset, the extracted concepts include: *clothing, feathers, flowers, forest, liquid, ocean waves, tennis ball, whiskers*, and many more. These concepts enhance our method's ability to differentiate between InD and OOD samples by focusing on fine-grained, interpretable features.

### A.1.2 Concepts Filtering

Concepts filtering is a critical step in our framework to ensure that the extracted concepts are both semantically meaningful and relevant to the tasks of multi-label OOD detection. This process removes noisy, redundant, or overly generic concepts to refine the final set of usable concepts. Following the method in LF-CBM Oikarinen et al. (2023), our filtering pipeline includes four main stages, as detailed below:

**1. Aggregation of Initial Concepts.** The concepts are initially collected from three semantic sources: *important features*, *superclasses*, and *common surroundings*, which are extracted using the prompts described in Section A.1.1. The aggregated set of concepts contains raw candidate phrases that may include overlaps, generic terms, or phrases irrelevant to the task.

At this stage, the aggregated set of concepts may contain noisy terms such as overly lengthy descriptions, overly generic phrases (e.g., "forest," "a barn"), or duplicates. To address these issues, we apply the following filtering stages.

**2. Length-Based Filtering.** Concepts exceeding a predefined maximum character length (`MAX_LEN`) are removed. This helps eliminate verbose descriptions and overly detailed phrases that are unlikely to be useful for OOD detection. For example, sample concepts too long to use:

```
attached passenger or cargo cars
facial features: eyes, nose, mouth
bright orange and yellow flames
```

This step ensures that only concise, interpretable concepts remain.

**3. Class Similarity Filtering.** Concepts that are overly similar to any of the defined class labels are removed based on cosine similarity, with a threshold defined by `CLASS_SIM_CUTOFF`. This ensures that the retained concepts provide additional semantic value rather than simply repeating class names. For example, sample outputs:

```
Class:airport - Deleting airport
Class:beach - Deleting a beach
Class:birds - Deleting birds
```

By removing concepts that are too similar to existing class labels, we ensure that the filtered concepts provide new, complementary information.

**4. Concept-to-Concept Similarity Filtering.** Finally, concepts that are overly similar to one another are removed using a threshold defined by `OTHER_SIM_CUTOFF`. This step eliminates redundancy by ensuring that only distinct concepts remain. Cosine similarity is used to measure the closeness between concepts.

Sample outputs:

```
Forest - forest, sim:0.9990
    - Deleting forest
a badge - badge, sim:0.9378
    - Deleting badge
a barn - a barn, sim:1.0000
    - Deleting a barn
a glass - a glass, sim:1.0000
    - Deleting a glass
```

This step ensures that the final set of concepts is diverse and avoids unnecessary overlaps, improving the semantic granularity of the filtered concepts.

**Summary of Filtering Process.** After these steps, the concept set is significantly refined, leaving only concise, semantically rich, and diverse concepts. This rigorous filtering pipeline is crucial for maintaining high interpretability and robustness in our OOD detection framework, as it ensures that the remaining concepts contribute meaningfully to the decision-making process.

### A.1.3 Concept Examples

The following are examples of concepts extracted from the COCO2017 dataset: *clothing, cushions, feathers, flowers, forest, liquid, ocean waves, sand, sidewalk, snow, soil, tracks, whiskers, wine, alarm button, camera lens, clouds, etc.* And the examples of concepts extracted from the Pascal VOC dataset: *a bicycle bell, a bicycle light, cat food, charger, clawed paws, claws, clouds, cockpit windows, flowers, food, four legs, four rubber tires, turbine, turbojet engines, two arms, two ears, two eyes, two large horns, two round wheels, various input ports, vertebrate, vessel, wagging tail, wallet, watch, etc.*

These concepts span a diverse set of categories, reflecting the dataset's complexity and our method's ability to capture fine-grained details relevant to OOD detection.

This process ensures that our model leverages rich, interpretable semantic information for robust and scalable OOD detection.

### A.2 Additional Experimental Results

### A.2.1 Details of the Experimental Setup

We conduct all experiments on NVIDIA 3090 GPUs using PyTorch 1.11.0, Python 3.8, and CUDA 11.3 on Ubuntu 20.04. Our experiments leverage both InD datasets and OOD datasets, as summarized in Table 2. Each dataset is processed to ensure no overlapping classes between InD and OOD datasets.

**Metrics Description.** The metrics used to evaluate OOD detection are:

- **AUROC**: Measures the area under the Receiver Operating Characteristic curve, providing an overall assessment of the separability between ID and OOD samples. Higher AUROC values indicate better detection performance.

- **FPR@95**: Represents the False Positive Rate when the True Positive Rate is fixed at 95%. A lower FPR@95 indicates fewer OOD samples being incorrectly classified as ID samples.

**Computational Cost.** Our method is designed to be computationally efficient. Generating concepts and embeddings requires minimal additional resources, and the measured runtime for OOD detection is 4.93 ms per image (203 images/s) on CLIP-B/16 with batch size 8 on an NVIDIA L20 GPU (Appendix A.2.4). The concept generation phase uses the target InD labels once; using GPT-4 prompts costs less than $5 even for datasets with 1000 classes. This efficiency ensures scalability for large-scale datasets without compromising performance.

### A.2.2 Implementation Details

In this section, we provide details about the implementation and parameter settings for our OOD detection framework. The design ensures computational efficiency while leveraging semantic concepts for improved interpretability.

**Feature Extraction.** Our framework extracts image features using a pre-trained vision-language model. These features are normalized to unit norm for consistency and then compared with precomputed concept embeddings using cosine similarity. This comparison generates three similarity scores:

- **Concept Similarity:** Measures the alignment between the image and semantic concepts.

- **Positive Similarity:** Measures the alignment with positive class-specific textual features.

- **Negative Similarity:** Measures the alignment with negative textual features, representing irrelevant or OOD characteristics.

**Scoring Mechanism.** The OOD detection score is computed using the top-k similarities of positive, negative, and concept features. The following steps outline the scoring mechanism:

- **Top-k Selection:** For each image, the top-k highest similarity scores are selected for positive, negative, and concept features. The value of $k$ is controlled by the `ood_topk` parameter, and empirically determined values are used for different datasets. For example, in ImageNet, the predefined settings are $k_{\text{pos}} = 1$, $k_{\text{concept}} = 500$, and $k_{\text{neg}} = 500$.

- **Score Computation:** The score is calculated based on the sampling method:
  - **Mean-based:** Computes the difference between the means of the top-k positive and negative similarities, adjusted by the mean of concept similarities.
  - **Median-based:** Replaces mean with median for robustness against outliers.
  - **Sum-based:** Uses the sum of top-k similarities instead of the mean.
  - **Percentile-based:** Computes scores based on a specified percentile (e.g., 90th percentile) of the top-k similarities.
  - **EMD-based:** Calculates Earth Mover's Distance (EMD) to evaluate the distributional differences between positive, negative, and concept similarities.

The default sampling method is **mean-based**, as it balances computational efficiency with robust performance.

**Hyperparameters and Sampling Strategies.** The hyperparameters used in our framework are summarized in Table 6. These settings were chosen to optimize performance across multiple datasets while maintaining computational efficiency.

Table 6: Optimal Hyperparameters for OOD Detection

| Parameter | Value |
|---|---|
| `ood_topk` (Pascal VOC) | 0.2 |
| `ood_topk` (MS-COCO) | 0.05 |
| `pos_k` (ImageNet InD) | 1 |
| `concepts_k` (ImageNet InD) | 500 |
| `neg_k` (ImageNet InD) | 500 |
| `sample_method` | mean |

**Scalability and Adaptability.** The scoring mechanism and top-k selection are adaptable to various datasets and tasks. For example, for single-label datasets such as ImageNet, lower $k_{\text{pos}}$ values are optimal, while for complex multi-label datasets, larger $k_{\text{concept}}$ and $k_{\text{neg}}$ values improve detection robustness.

### A.2.3 Detailed Results

In this section, we present additional results and analyses of our method, covering ablation studies, model architecture comparisons, distance metrics, prompt variations, and performance on ImageNet ID and OOD datasets.

**Ablation Study on Score Components.** Table 7 evaluates the individual and combined contributions of score components $S_{\text{A}}$ and $S_{\text{B}}$ in our OOD detection framework. We observe the following:

- The combined use of $S_{\text{A}}$ and $S_{\text{B}}$ consistently achieves the best performance across all metrics, highlighting their complementary benefits.

Table 7: Ablation Study Results for Multi-Label OOD Detection on Pascal VOC and COCO ID Datasets

| Top-k | $S_A$ | $S_B$ | Pascal VOC ID | | | COCO ID | | |
|---|---|---|---|---|---|---|---|---|
| | | | FPR95 | AUROC | AUPR | FPR95 | AUROC | AUPR |
| 0.05 | ✗ | ✓ | 39.39 | 89.08 | 86.59 | 26.65 | 94.40 | 97.17 |
| 0.1 | ✗ | ✓ | 35.59 | 91.10 | 89.52 | 37.20 | 92.27 | 96.07 |
| 0.2 | ✗ | ✓ | 36.12 | 91.72 | 90.48 | 48.87 | 89.49 | 94.68 |
| 0.3 | ✗ | ✓ | 38.92 | 90.98 | 89.57 | 53.80 | 87.88 | 93.87 |
| 0.5 | ✗ | ✓ | 43.39 | 89.34 | 87.57 | 58.97 | 86.13 | 92.95 |
| 0.8 | ✗ | ✓ | 47.04 | 87.53 | 85.09 | 62.90 | 84.61 | 92.10 |
| 1 | ✗ | ✓ | 49.40 | 86.72 | 83.92 | 64.46 | 83.71 | 91.59 |
| 0.05 | ✓ | ✗ | 45.79 | 90.13 | 90.06 | 28.88 | 92.39 | 95.67 |
| 0.1 | ✓ | ✗ | 37.55 | 91.84 | 91.23 | 30.32 | 91.47 | 94.95 |
| 0.2 | ✓ | ✗ | 36.43 | 91.73 | 90.69 | 33.70 | 90.09 | 94.04 |
| 0.3 | ✓ | ✗ | 38.66 | 90.43 | 88.84 | 37.29 | 88.47 | 92.99 |
| 0.5 | ✓ | ✗ | 42.60 | 87.34 | 84.37 | 43.17 | 85.12 | 90.66 |
| 0.8 | ✓ | ✗ | 47.37 | 83.73 | 79.03 | 48.80 | 81.08 | 87.57 |
| 1 | ✓ | ✗ | 50.95 | 81.88 | 76.37 | 52.62 | 79.00 | 85.89 |
| 0.05 | ✓ | ✓ | 36.28 | 91.28 | 90.45 | 20.37 | 95.07 | 97.32 |
| 0.1 | ✓ | ✓ | 28.07 | 93.60 | 92.68 | 24.16 | 93.91 | 96.60 |
| 0.2 | ✓ | ✓ | 24.78 | 94.27 | 93.13 | 29.08 | 92.39 | 95.70 |
| 0.3 | ✓ | ✓ | 26.76 | 93.62 | 92.26 | 32.95 | 91.02 | 94.88 |
| 0.5 | ✓ | ✓ | 32.23 | 91.82 | 89.95 | 39.61 | 88.48 | 93.30 |
| 0.8 | ✓ | ✓ | 38.84 | 89.31 | 86.72 | 46.57 | 85.20 | 91.10 |
| 1 | ✓ | ✓ | 43.28 | 87.75 | 84.74 | 50.54 | 83.26 | 89.75 |

- Concepts provide significant semantic richness, which substantially enhances OOD detection accuracy, as evidenced by the improvements in AUROC and FPR95 when both scores are used together.

Table 8: Performance Comparison of Different Backbone Models

| Distance | Architecture | Pascal VOC | | | | COCO | | | | ImageNet | |
|---|---|---|---|---|---|---|---|---|---|---|---|
| | | ImageNet22k | | Textures | | ImageNet22k | | Textures | | Textures | |
| | | FPR95 | AUROC | FPR95 | AUROC | FPR95 | AUROC | FPR95 | AUROC | FPR95 | AUROC |
| | ResNet50 | 25.28 | 93.32 | 8.76 | 97.79 | 26.83 | 93.14 | 10.53 | 97.17 | 43.72 | 91.68 |
| | ResNet50x16 | 28.69 | 93.88 | 18.54 | 95.89 | 18.41 | 95.88 | 13.96 | 96.52 | 42.28 | 89.34 |
| Mean | ViT-B/32 | 28.91 | 92.59 | 22.76 | 95.23 | 25.00 | 93.60 | 15.68 | 96.18 | 38.46 | 91.12 |
| | ViT-B/16 | 23.87 | 94.32 | 21.58 | 95.14 | 20.37 | 95.07 | 21.63 | 94.53 | 39.41 | 91.10 |
| | ViT-L/14 | 40.08 | 91.00 | 24.28 | 94.55 | 23.52 | 95.24 | 22.71 | 94.25 | 57.84 | 84.46 |

**Model Architecture Comparison.** Table 8 compares the performance of different backbone architectures, including ResNet-based and ViT-based CLIP models. Key observations include:

- ViT-B/16 consistently achieves the best performance on most datasets, particularly in AUROC, demonstrating its superior ability to capture fine-grained semantic relationships.

- ResNet50 provides competitive results on simpler datasets like Textures, showcasing its efficiency and robustness in less complex scenarios.

- ViT-L/14 suffers from performance drops in some cases due to overfitting or its high complexity, which may not align with our zero-shot setup.

**Comparison of Distance Metrics.** Table 9 evaluates the impact of different distance metrics, including mean, median, percentile, and EMD, on OOD detection. The key findings are as follows:

Table 9: Performance Comparison Using Different Distance Metrics

| Architecture | Distance Metric | Pascal VOC | | | | COCO | | | | ImageNet | |
| | | ImageNet22k | | Textures | | ImageNet22k | | Textures | | Textures | |
| | | FPR95 | AUROC | FPR95 | AUROC | FPR95 | AUROC | FPR95 | AUROC | FPR95 | AUROC |
|---|---|---|---|---|---|---|---|---|---|---|---|
| ViT-B/16 | Percentile(0.5) | 35.81 | 91.05 | 42.23 | 90.50 | 31.83 | 92.31 | 35.10 | 91.87 | 56.32 | 84.97 |
| | Percentile(0.75) | 27.15 | 93.26 | 26.62 | 93.99 | 22.83 | 94.66 | 23.15 | 94.29 | 53.84 | 85.81 |
| | Percentile(0.9) | 29.83 | 93.03 | 22.46 | 95.38 | 28.19 | 94.09 | 24.35 | 93.70 | 50.56 | 87.34 |
| | Median | 38.19 | 90.09 | 48.54 | 88.61 | 38.35 | 90.48 | 43.14 | 89.73 | 56.80 | 84.78 |
| | EMD | 33.50 | 92.61 | 18.25 | 95.85 | 99.63 | 25.25 | 91.88 | 50.02 | 99.77 | 30.32 |
| | Mean | 23.87 | 94.32 | 21.58 | 95.14 | 20.37 | 95.07 | 21.63 | 94.53 | 39.41 | 91.10 |
| ResNet50 | Percentile(0.5) | 34.40 | 90.17 | 18.42 | 96.00 | 34.01 | 91.08 | 16.23 | 96.18 | 35.90 | 91.57 |
| | Percentile(0.75) | 27.19 | 92.35 | 9.44 | 97.62 | 28.95 | 92.84 | 12.02 | 97.02 | 35.97 | 91.55 |
| | Percentile(0.9) | 29.68 | 92.59 | 8.99 | 97.85 | 29.87 | 92.77 | 13.73 | 96.32 | 35.97 | 91.46 |
| | Median | 38.09 | 89.07 | 28.43 | 94.41 | 38.30 | 89.52 | 23.74 | 94.95 | 35.94 | 91.57 |
| | EMD | 30.91 | 92.98 | 8.56 | 97.92 | 91.03 | 56.06 | 44.27 | 85.19 | 79.47 | 76.13 |
| | Mean | 25.28 | 93.32 | 8.76 | 97.79 | 26.83 | 93.14 | 10.53 | 97.17 | 43.72 | 91.68 |

- The mean-based scoring strategy consistently achieves stable and optimal performance across all datasets.

- Percentile-based metrics perform well in specific scenarios, such as higher percentiles (e.g., 90th) on complex datasets, but lack the overall robustness of the mean.

- EMD, while theoretically sound, exhibits higher computational costs and instability in some cases, making it less practical for large-scale OOD detection.

Table 10: Performance Comparison of Different Prompts from Jiang et al. (2024)

| Prompt | Pascal VOC | | | | COCO | | | | ImageNet | |
| | ImageNet22k | | Textures | | ImageNet22k | | Textures | | Textures | |
| | FPR95 | AUROC | FPR95 | AUROC | FPR95 | AUROC | FPR95 | AUROC | FPR95 | AUROC |
|---|---|---|---|---|---|---|---|---|---|---|
| the {label} | 23.87 | 94.32 | 21.58 | 95.14 | 20.37 | 95.07 | 21.63 | 94.53 | 39.41 | 91.10 |
| A photo of a {label}. | 73.23 | 79.83 | 76.94 | 81.12 | 48.30 | 86.21 | 62.04 | 87.95 | 58.25 | 85.19 |
| A good photo of a {label}. | 67.50 | 81.32 | 51.03 | 89.82 | 49.24 | 86.68 | 39.42 | 92.47 | 47.90 | 88.30 |
| The nice {label}. | 47.83 | 88.88 | 32.11 | 93.78 | 34.49 | 91.19 | 31.97 | 94.08 | 41.09 | 90.78 |
| A blurry photo of a {label}. | 73.60 | 78.41 | 49.67 | 89.82 | 63.49 | 82.65 | 27.82 | 94.72 | 41.44 | 90.62 |
| A cropped photo of a {label}. | 91.56 | 64.90 | 97.57 | 51.82 | 78.22 | 73.28 | 94.21 | 71.14 | 75.74 | 78.39 |

**Effect of Prompt Variations.** Table 10 analyzes the impact of various prompt designs on our framework. Our observations include:

- Short and direct prompts, such as `the {label}`, yield the best performance, with significantly higher AUROC and lower FPR95.

- Additional prompts, such as `A photo of a {label}` or `A cropped photo of a {label}`, degrade performance. This is likely due to the inclusion of unnecessary context, which dilutes the semantic alignment between image features and concepts.

- The results suggest that concise prompts are essential for effectively representing concepts and tags in our framework.

**Performance on ImageNet ID and OOD Datasets.** Table 11 compares our method with existing approaches on ImageNet ID and multiple OOD datasets in single-label OOD detection tasks. Although originally designed for multi-label scenarios, our method demonstrates strong competitiveness in single-label OOD detection tasks, achieving state-of-the-art (SoTA) performance on certain datasets and competitive results across most others.

**Summary.** These detailed results demonstrate the effectiveness and adaptability of our method across various scenarios and configurations. Our combined scoring mechanism, robust backbone architectures, and

Table 11: Performance comparison on ImageNet ID and various OOD datasets. Metrics: FPR95 (%) and AUROC (%).

| Method | Textures | | iNaturalist | | SUN | | Places | |
|---|---|---|---|---|---|---|---|---|
| | FPR95 | AUROC | FPR95 | AUROC | FPR95 | AUROC | FPR95 | AUROC |
| **CLIP with Training/ Fine-tuning** | | | | | | | | |
| Energy Ming et al. (2022) | 51.18 | 88.09 | 21.59 | 95.99 | 34.28 | 93.15 | 36.64 | 93.01 |
| MSPMing et al. (2022) | 64.96 | 78.16 | 40.89 | 88.63 | 65.81 | 81.24 | 67.90 | 80.14 |
| MCM-SeTAR-FT Li et al. (2024b) | 53.32 | 87.72 | 32.95 | 93.41 | 30.26 | 93.81 | 38.56 | 91.24 |
| GL-MCM-SeTAR-FTLi et al. (2024b) | 51.18 | 87.01 | 21.62 | 95.43 | 23.38 | 94.89 | 32.60 | 91.93 |
| NPOSJiang et al. (2024) | 46.12 | 88.80 | 16.58 | 96.19 | 43.77 | 90.44 | 45.27 | 89.44 |
| **CLIP without Training** | | | | | | | | |
| MCMMing et al. (2022) | 57.77 | 86.11 | 30.91 | 94.61 | 37.59 | 92.57 | 44.69 | 89.77 |
| GL-MCMLi et al. (2024b) | 57.41 | 83.73 | 15.34 | 96.62 | 30.65 | 93.01 | 37.76 | 90.07 |
| MCM+SeTARLi et al. (2024b) | 55.83 | 86.58 | 26.92 | 94.67 | 35.57 | 92.79 | 42.64 | 90.16 |
| GL-MCM+SeTARLi et al. (2024b) | 54.17 | 84.59 | 13.36 | 96.92 | 28.17 | 93.36 | 36.80 | 90.40 |
| CLIPNJiang et al. (2024) | 40.83 | 90.93 | 23.94 | 95.27 | 26.17 | 93.93 | 33.45 | 92.28 |
| NegLabelJiang et al. (2024) | 43.56 | 90.22 | 1.91 | 99.49 | 20.53 | 95.49 | 35.59 | 91.64 |
| **Ours (without Training, Zero-shot)** | | | | | | | | |
| Ours-RN50 | 43.72 | 91.68 | 7.48 | 98.39 | 60.15 | 87.40 | 66.45 | 83.37 |
| Ours-ViT-B/16 | 39.41 | 91.10 | 6.03 | 98.69 | 37.11 | 92.47 | 43.84 | 89.52 |

carefully designed prompts ensure consistent performance improvements, making our approach a reliable choice for OOD detection tasks.

**Additional Analysis On Single-label Dataset** We also look at the SUN and Places datasets in the study. Both datasets represent scenes rather than object-centric categories, which dilutes the semantic contrast leveraged by CLIP-based embeddings. In the SUN dataset, large rocks, wilderness and blue sky often appear. In the Places dataset, objects included in the ID dataset, such as people and cars, often appear in the background by mistake. This leads to the performance degradation of the OOD detection method based on concept design. Fig 6 presents samples from both datasets.

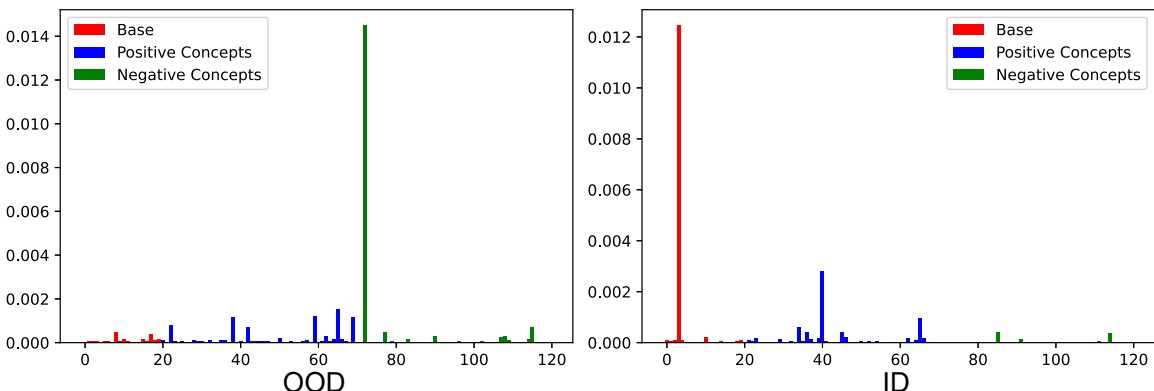

Figure 5: Similarities of 3 types of concepts

**Multi-Label ID/OOD and Baseline Comparison** We run our experiment under setting where both ID and OOD datasets are multi-label (VOC+O365), and also comapre CMOOD with YolOOD (Zolfi et al., 2024), which utilizes object detection concepts for multi-label OOD detection. According to Tab. 12,

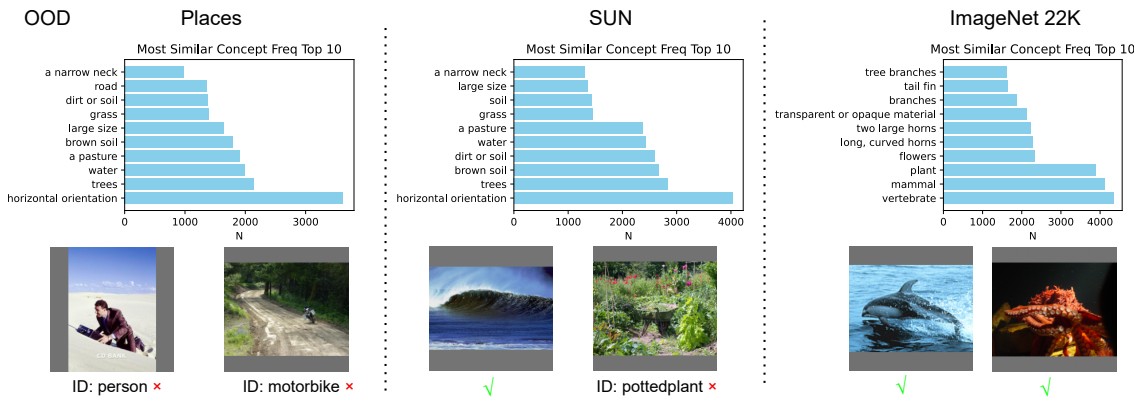

Figure 6: Concepts of Single-label Datasets

CMOOD outperforms mainstream advanced methods under zero-shot, box-free settings. Although YolOOD performs well on multi-label OOD datasets like Objects-365, it assumes access to bounding boxes and trains a full detector, which is not our setting.

Table 12: YolOOD under comparable settings. O365_out for Objects-365_out dataset, which is a multilabel dataset

| Method | ID: VOC OOD: O365_out | ID: VOC OOD: IN22K | ID: COCO OOD: O365_out | ID: COCO OOD: IN22K |
|---|---|---|---|---|
| *Note:* **YolOOD** uses *training + box annotations*, while all others are *zero-shot, image-only* methods. | | | | |
| **YolOOD** (trained + boxes) | **16.38 / 96.60** | 38.85 / 91.03 | **11.53 / 97.29** | 34.79 / 90.84 |
| **MCM** (zero-shot) | 83.09 / 80.01 | 70.81 / 80.37 | 70.81 / 84.65 | 63.34 / 86.10 |
| **NegLabel** (zero-shot) | 80.71 / 79.76 | 35.83 / 91.18 | 71.27 / 81.68 | 33.24 / 90.19 |
| **Ours (CMOOD)** (zero-shot) | 56.72 / 86.30 | **23.87 / 94.32** | 56.16 / 84.23 | **20.37 / 95.07** |

### A.2.4 Inference Efficiency and FLOPs

We benchmark inference efficiency on Pascal VOC using the CLIP ViT-B/16 backbone with batch size 8 on an NVIDIA L20 GPU (three runs after five warm-up batches). Metrics include total time, throughput, per-image latency, and GPU memory. CMOOD reaches 203.06 images/second (4.93 ms/img) with 1565 MB memory, outperforming all baselines by 28–44% while using less memory. Table 13 summarizes the results. We also report theoretical FLOPs for the scoring computation (encoder FLOPs excluded); despite higher FLOPs than simple scores, CMOOD remains fastest because its matrix multiplications parallelize efficiently (Table 14).

Table 13: Inference efficiency on Pascal VOC with ViT-B/16 (batch size 8, NVIDIA L20). Mean ± std over three runs.

| Method | Time (s) | Images/s | ms/img | GPU (MB) |
|---|---|---|---|---|
| **CMOOD (ours)** | $0.25 \pm 0.01$ | **$203.06 \pm 6.99$** | **$4.93 \pm 0.17$** | **$1565 \pm 0$** |
| NegLabel | $0.34 \pm 0.02$ | $146.28 \pm 8.64$ | $6.86 \pm 0.41$ | $1807 \pm 0$ |
| Energy | $0.34 \pm 0.02$ | $148.42 \pm 7.80$ | $6.76 \pm 0.35$ | $1807 \pm 0$ |
| MSP | $0.35 \pm 0.03$ | $144.41 \pm 11.75$ | $6.97 \pm 0.59$ | $1807 \pm 0$ |
| MaxLogit | $0.35 \pm 0.01$ | $141.40 \pm 3.86$ | $7.08 \pm 0.20$ | $1807 \pm 0$ |

Table 14: FLOPs for OOD scoring computation (shared encoder excluded).

| Method | FLOPs/Image | GFLOPs |
|---|---|---|
| **CMOOD (ours)** | 1,081,794 | 0.001 |
| NegLabel | 749,256 | 0.001 |
| Energy | 20,540 | 0.000 |
| MSP | 20,540 | 0.000 |
| MaxLogit | 20,540 | 0.000 |

### A.2.5  Hyperparameter Robustness and Zero-Shot Configuration

We assess sensitivity to key design choices without validation tuning. Varying the Top-$k$ parameter in $[0.05, 0.2]$ changes AUROC by at most 3.0% on Pascal VOC and 2.7% on COCO, so we fix Top-$k$ in this range across datasets. Alternative aggregation metrics (mean, percentile$_{0.75}$, percentile$_{0.9}$, median) all exceed 90% AUROC on Pascal VOC, indicating low sensitivity to similarity aggregation. Prompt templates remain stable, with the simple "the {label}" form outperforming more complex phrases. These choices keep the method zero-shot (no gradient updates or validation-based tuning).

### A.2.6  Partial OOD Behavior

To test mixed images containing both ID and unseen objects, we resplit Pascal VOC into 11 ID classes (aeroplane, bicycle, bird, boat, bottle, bus, car, cat, person, dog, cow) and 9 OOD classes (chair, diningtable, horse, motorbike, pottedplant, sheep, sofa, train, tvmonitor). Without retraining, we score 500 validation images grouped by ID/OOD content. Scores degrade smoothly from pure ID to pure OOD, showing image-level aggregation handles mixed scenes (Table 15).

Table 15: CMOOD scores on partial OOD categories in Pascal VOC.

| Category | ID Ratio | Mean | Std | Count |
|---|---|---|---|---|
| Pure ID | 100% | 3.15 | 2.55 | 367 |
| Mixed | 25–75% | 1.92 | 2.18 | 85 |
| Pure OOD | 0% | -0.20 | 3.00 | 48 |

### A.2.7  Concept Set Sizes and Bias Mitigation

Table 16 lists positive concept counts; they scale with dataset size while averaging 7–16 concepts per label. During generation we filter by class similarity and inter-concept similarity to remove near-duplicates or loosely related surroundings, and the Top-$k$ score uses only the most similar concepts. Negative concepts provide a counter-signal that mitigates residual background correlations.

Table 16: Positive concept set sizes across datasets.

| Dataset | # Base Labels | # Positive Concepts | Avg./Label |
|---|---|---|---|
| Pascal VOC | 20 | 317 | 15.9 |
| COCO | 80 | 1060 | 13.3 |
| ImageNet | 1000 | 7683 | 7.7 |

### A.2.8  ID vs. OOD Samples

The Fig. 7 and Fig. 8 showcases example images paired with their respective bar plots, highlighting the concept weights for each image. In these bar plots, positive concepts are depicted in red, while negative concepts are shown in blue. This visualization provides a clear understanding of the semantic features contributing to the OOD score, facilitating the interpretation of why specific samples are identified as ID or OOD.

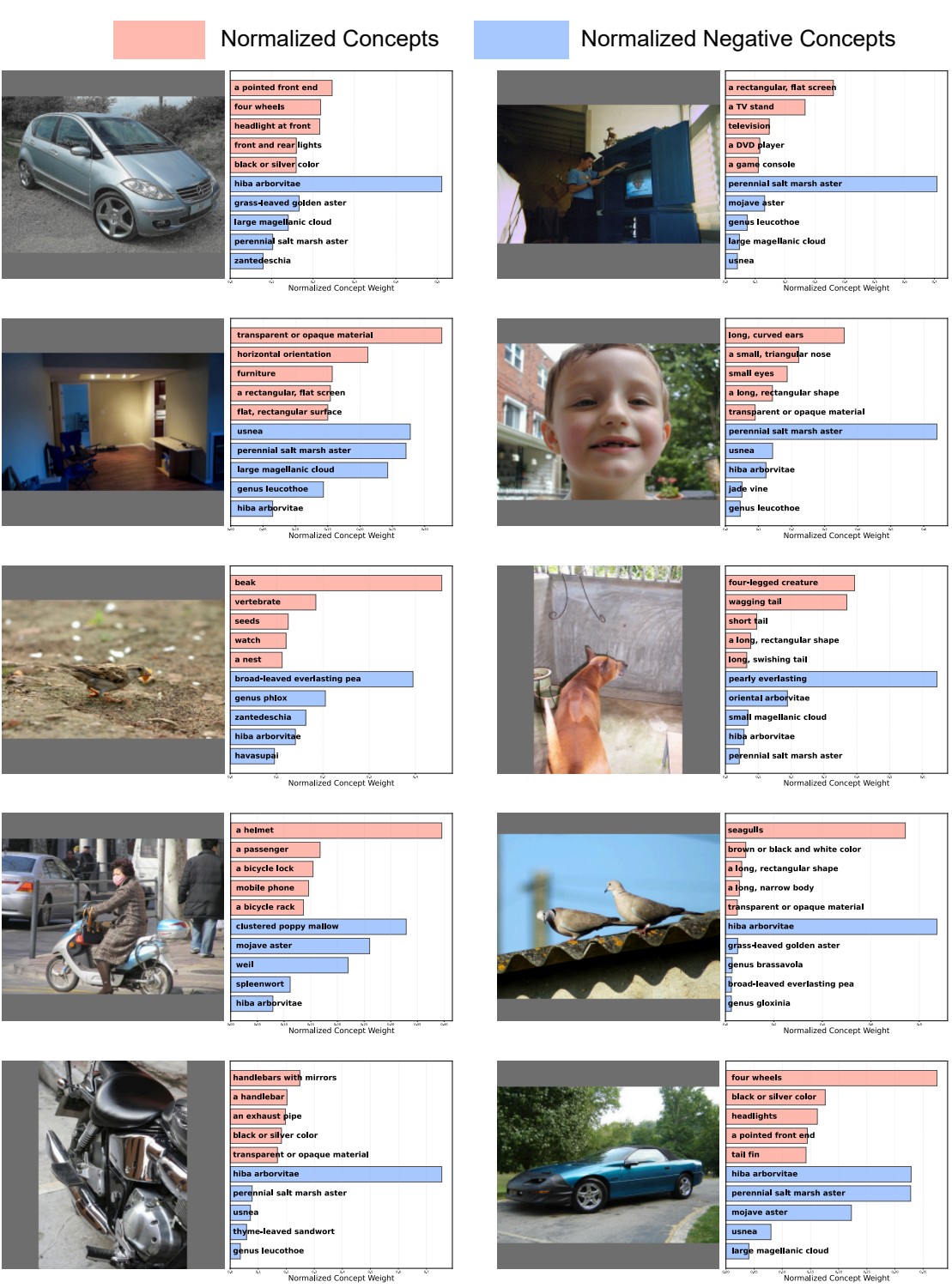

Figure 7: Some examples of in-distribution samples. The weights have been normalized to the five largest values for easy observation.

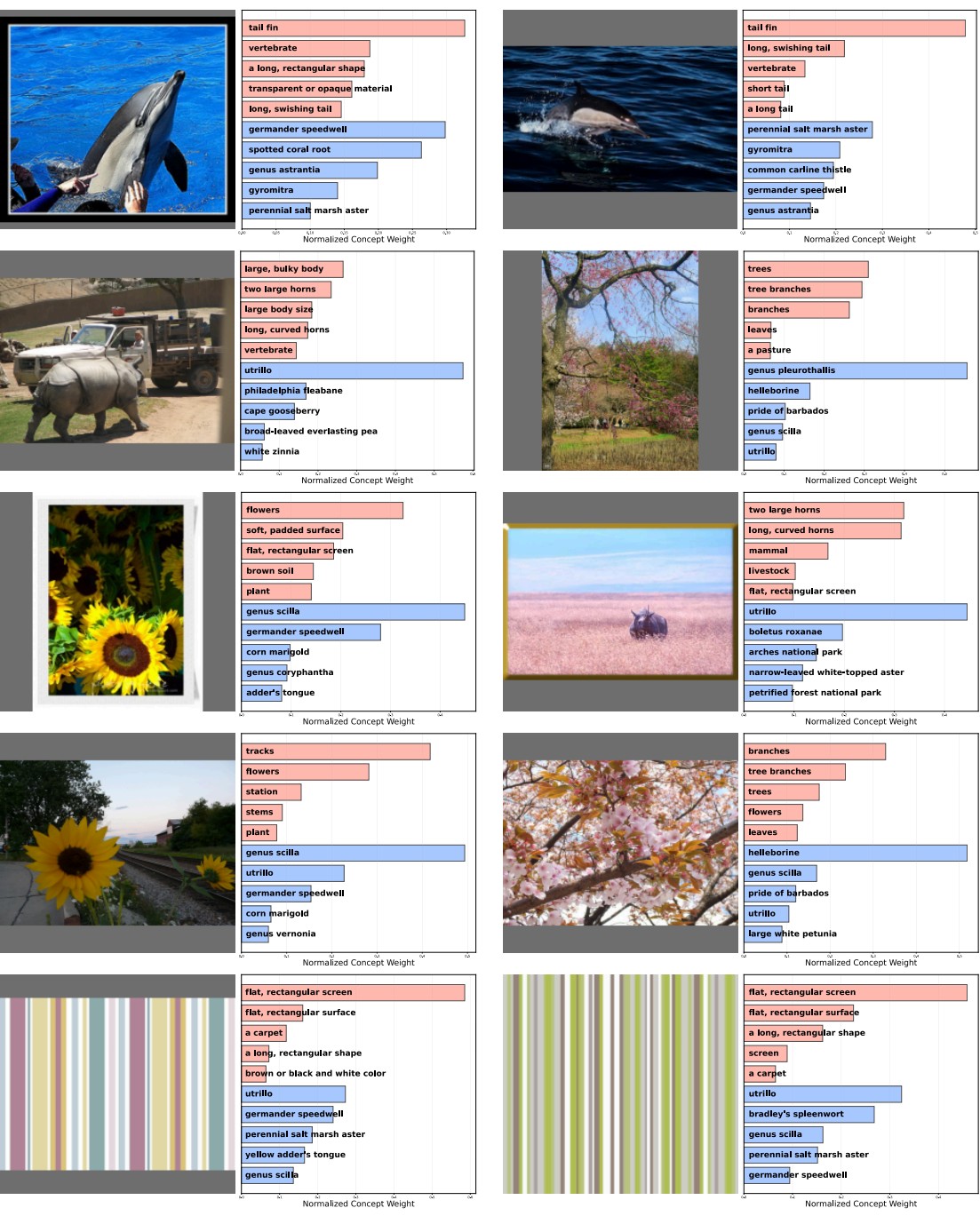

Figure 8: Some examples of out-of-distribution samples. The weights have been normalized to the five largest values for easy observation.

