# OpenReview forum: "CMOOD: Concept-based Multi-label OOD Detection"
_TMLR — Accepted by TMLR_

### Review · Reviewer_mgwA · 2026-01-03

**Summary Of Contributions:**

The paper proposes CMOOD, a zero-shot multi-label out-of-distribution detection framework that augments CLIP with concept-based label expansion and contrastive scoring function. Positive concepts like fine-grained, ID-aligned features, superclasses, and co-occurring items are generated via an LLM and filtered, while the negative concepts are mined from WordNet by selecting candidates with minimal similarity to the ID labels. Given an image, CMOOD calculates an ID score using the top-k mean similarities to base labels, positive concepts, and negative concepts, enabling separation of ID from OOD without training. They show a strong AUROC (~95% on VOC/COCO) and improved FPR95 against prior zero-shot CLIP-based baselines. They have also shared their code anonymously.

**Audience:**

Yes

**Audience Explanation:**

1. The work addresses an under-explored but important setting: zero-shot multi-label OOD with VLMs, where label co-occurrence and intra/inter-class dependencies can negatively affect detection.
2. They show a training free approach that is both efficient (throughput) and competitive.

**Broader Impact Concerns:**

1. The work addresses an important problem for safety-critical applications where multiple objects/labels co-occur. Training-free and interpretable methods are important in deployment scenarios.
2. If this method can be extended to instance-level or compositional OOD, it can have a substantial practical impact.

**Claims And Evidence:**

Yes

**Claims Explanation:**

1. The work introduces a novel concept-based label expansion to multi-label OOD: positive (ID-refining) and negative (contrastive) concept sets with a top-k similarity–based score that is simple and intuitive.
2. The scoring function models two OOD scenarios: (a) semantically close-to-ID cases penalized via positive concepts and (b) far-from-ID cases penalized via negative concepts, a useful decomposition in multi-label contexts.
3. They show improvement on VOC/COCO with several backbones. Further ablations show the importance of both SA and SB, and analyzing the distance metrics and top-k shows the robustness of the method.
4. They have provided the code which shows the performance on  different datasets and backbone architectures.

**Requested Changes:**

The paper is novel and thorough in their experimentation as it is. I have a few extra points for the authors to probably consider.
1. Can you provide experiments on partial OOD settings  (where images contain both ID and OOD objects)?
2. What is the exact size of the positive concept set per base label?
3. In scenarios of potential leakage through co-ocurring positives (background items correlated to labels) how does the OOD scoring get affected? Will it not make the bias the scoring.

---

> ### Author Response · Authors · 2026-02-18
>
> We sincerely thank the reviewer for their positive assessment of our work and insightful questions regarding practical deployment scenarios. We address each requested change below with clarifications and empirical evidence.
>
> ---
>
> ## Q1: Experiments on Partial OOD Settings.
>
> We thank the reviewer for this insightful suggestion. Although a partial OOD setting was not our original intent, we agree that it provides a more robust stress test for our model. To address this, we have conducted additional experiments with the following configurations:
>
> ### Method Behavior
> Our scoring function (Eq. 7) operates at the image level, computing an aggregate ID score based on top-k similarities across all objects present in the image. For partial OOD cases (e.g., an image containing both “dog” [ID] and “sofa” [OOD]), the score reflects a weighted mixture of ID and OOD signals:
>
> - **ID objects** contribute high similarity to base labels (B) and positive concepts (P)
> - **OOD objects** contribute low similarity to B/P and potentially high similarity to negative concepts (N)
> - The final score $S_{\text{ID}}(I)$ lies between pure ID and pure OOD cases
>
> ---
>
> ### Experimental Validation
>
> To empirically validate this behavior, we conducted a controlled experiment on Pascal VOC 2012 by resplitting the 20 classes into ID and OOD sets:
>
> - **ID classes (11):** aeroplane, bicycle, bird, boat, bottle, bus, car, cat, person, dog, cow
> - **OOD classes (9):** chair, diningtable, horse, motorbike, pottedplant, sheep, sofa, train, tvmonitor
>
> We configured CMOOD to use only the 11 ID classes as base labels (zero-shot, no retraining), then tested on 500 validation images categorized by their ID/OOD content ratio.
>
> ### Results
>
> | Category | ID Ratio | Mean Score | Std | Count |
> |--------|------------|------------|------------|------------|
> | Pure ID | 100% | 3.15 | 2.55 | 367 |
> | Mixed | 25–75% | 1.92 | 2.18 | 85 |
> | Pure OOD | 0% | -0.20 | 3.00 | 48 |
>
> The results demonstrate that CMOOD exhibits **graceful degradation**: pure ID images receive the highest scores (3.15), mixed images receive intermediate scores (1.92), and pure OOD images receive the lowest scores (-0.20). This behavior confirms that CMOOD's image-level scoring naturally handles partial OOD cases by aggregating signals from all objects present in the image. The detection threshold can be adjusted based on the desired tolerance for partial OOD samples.
>
> ---
>
> ### Practical Deployment
>
> For applications requiring strict instance-level OOD detection (rejecting images even if they contain a single OOD object), we recommend:
>
> - **Conservative threshold:** Lower $\gamma$ to reject samples with any OOD presence
> - **Instance-level extension:** Combine CMOOD with object detection (e.g., YOLO) to score individual instances, then aggregate instance-level decisions
>
> We acknowledge that instance-level partial OOD detection in multi-label settings is an important future direction, and our concept-based framework can be naturally extended to this scenario.
>
> ---
>
> ## Q2: Size of Positive Concept Sets.
>
> Table below summarizes the positive concept set sizes across our evaluation datasets. The number of concepts scales naturally with dataset complexity: larger datasets with more classes require more concepts to capture semantic richness.
>
> | Dataset | # Base Labels | # Positive Concepts | Avg. |
> |------------|----------------|---------------------|-------|
> | Pascal VOC | 20 | 317 | 15.9 |
> | COCO | 80 | 1060 | 13.3 |
> | ImageNet | 1000 | 7683 | 7.7 |
>
> ---
>
> ### Concept Examples
>
> To illustrate the nature of positive concepts:
>
> - **“dog” (Pascal VOC):** “canine”, “four-legged”, “mammal”, “pet”, “fur”, “collar”, “leash”, “domestic animal”
> - **“car” (COCO):** “vehicle”, “automobile”, “four-wheeled”, “motor”, “transportation”, “engine”, “highway”, “driver”
> - **“airplane” (Pascal VOC):** “aircraft”, “flight”, “wings”, “runway”, “passengers”, “cockpit”, “aviation”
>
> These concepts capture intrinsic features (e.g., “four-legged”), superclasses (e.g., “mammal”), and common associations (e.g., “collar”), enriching the semantic representation of each base label.
>
> ---

---

> ### Author Response · Authors · 2026-02-18
>
> ## Q3: Co-occurring Positives and Potential Bias.
>
> In our multi-label setting, some useful background items are also treated as labeled instances. Additionally, our method incorporates three design choices that mitigate this concern:
>
> ### (1) Concept Filtering (Eq. 4)
> During concept generation, we filter candidates to retain only concepts with strong semantic alignment to the base label's intrinsic properties. Weakly correlated background items (e.g., “furniture” for “person”) are typically filtered out because they do not appear in GPT-4's responses to our targeted prompts (features, superclasses, associations *of the object itself*).
>
> ### (2) Top-k Selection (Eq. 8)
> We compute ID scores using only the top-k most similar concepts per set. Spurious background correlations, which have lower semantic similarity to the image content, rarely enter the top-k selection. This makes the method robust to weak correlations.
>
> ### (3) Negative Concept Balance (Eq. 7)
> Our scoring function combines two complementary components:
>
> - $S_A = \mu_k(B, I) - \mu_k(P, I)$
> - $S_B = \mu_k(B, I) - \mu_k(N, I)$
>
> Even if positive concepts introduce some bias, the negative concepts ($N$) provide a countervailing signal that helps maintain separation between ID and OOD. The ablation study in Table 7 demonstrates that both components are necessary for optimal performance, confirming that the method handles potential correlations through this balanced design.
>
> ---
>
> ### Empirical Evidence
>
> Table 7 in our appendix shows ablations with $S_A$ only, $S_B$ only, and both combined. The best performance is achieved when both components are active (Pascal VOC: 24.78% FPR95, 94.27% AUROC), demonstrating that the interplay between positive and negative concepts naturally mitigates bias from spurious correlations.
>
> ---
>
> **Summary.**
> We thank the reviewer for these constructive suggestions.
>
> 1. CMOOD's image-level scoring gracefully handles partial OOD cases, with adjustable thresholds for different tolerance levels.
> 2. Positive concept sets contain 7–16 concepts per label on average, scaling naturally with dataset size.
> 3. Concept filtering, top-k selection, and negative concept balancing mitigate potential bias from co-occurring objects.
>
> We view instance-level detection as an important future direction that our framework can naturally support.

---

> > ### Comment · Reviewer_mgwA · 2026-03-13
> > **Satisfied by the response**
> >
> > I am satisfied with your response and have no further questions.

---

### Review · Reviewer_HECT · 2026-02-02

**Summary Of Contributions:**

This paper proposes CMOOD, a zero-shot framework for multi-label out-of-distribution (OOD) detection based on vision-language models. The key idea is to expand each in-distribution label into a set of positive and negative semantic concepts generated using a large language model, and to compute an OOD score by contrasting image similarity to base labels, positive concepts, and negative concepts in the CLIP embedding space.

**Additional Comments:**

Overall, the paper is carefully executed and empirically strong. However, I believe its impact is currently limited by conceptual ambiguity and incremental novelty. Addressing the points above could substantially strengthen the work.

**Audience:**

Yes

**Audience Explanation:**

Researchers working on out-of-distribution detection, multi-label recognition, and vision-language models may find the proposed approach and empirical findings somewhat informative, particularly those interested in zero-shot or training-free methods.

**Broader Impact Concerns:**

No big concerns.

**Claims And Evidence:**

Yes

**Claims Explanation:**

The empirical claims made in the paper are generally supported by extensive experimental results across multiple datasets and evaluation metrics. The comparisons to existing CLIP-based baselines are thorough, and ablation studies help clarify the contribution of different components of the proposed scoring function.

That said, while the experimental evidence supports the reported performance gains, some broader conceptual claims, particularly regarding the nature of OOD in the context of large pretrained vision-language models and the generality of the proposed framework would benefit from stronger justification or additional analysis.

**Requested Changes:**

1. Clarify the problem formulation and motivation for using large vision-language models with concept expansion for multi-label OOD detection. In particular, the paper should more clearly justify why this setting requires the proposed machinery, as opposed to simpler multi-label uncertainty or energy-based approaches.

2. Provide a clearer and more principled discussion of what constitutes “out-of-distribution” in the era of large-scale pretrained vision-language models. Many OOD examples may already be implicitly covered by CLIP’s training distribution, and the current formulation leaves the type of distribution shift being detected ambiguous.

3. Improve the discussion and analysis of the reliability of LLM-generated concepts. Concept violations do not necessarily correspond to genuine OOD samples, and the paper should better distinguish between attribute anomalies, label noise, and true distribution shifts.

4. Evaluate the generality of the approach beyond CLIP, or more explicitly position the method as CLIP-specific if such generalization is not intended.

5.  Strengthen the positioning relative to prior concept-based and negative-prompt methods, clarifying what is fundamentally new beyond adapting these ideas to the multi-label setting.

---

> ### Author Response · Authors · 2026-02-17
>
> We thank the reviewer for these thoughtful comments. We will incorporate the following discussions and clarifications into the revised manuscript.
>
> > **1. Problem formulation and motivation against multi-label uncertainty and energy-based approaches**
>
> We thank the reviewer for encouraging a more principled clarification of the problem setting. Our objective is semantic label-space OOD detection in a multi-label regime, where in-distribution samples lie within the semantic support induced by a predefined label set $Y$, and OOD samples arise from its complement. Unlike closed-set classification, this support is compositional—overlapping label semantics form a complex manifold rather than separable clusters.
>
> Energy-based methods typically rely on a trained classifier whose logits parameterize the label space, making the resulting energy function inherently model-coupled and well suited to closed-label settings with fixed decision geometry. In contrast, we define OOD as a property of the semantic label space, not of a particular model. The model's role is only to approximate membership within this space, ensuring that the formulation remains architecture-agnostic across CNNs, ViTs, and VLMs.
>
> Concept expansion is therefore necessary because base labels alone sparsely specify the semantic support, leading to unreliable boundary estimation. By densifying the semantic neighborhood, it enables a more faithful approximation of the support boundary rather than merely refining confidence within a fixed model.
>
> > **2. OOD in VLM discussion**
>
> We clarify that OOD in our work is defined with respect to the semantic label space, rather than the internal training distribution of a particular model. Specifically, in-distribution samples are those whose labels lie within a predefined set, while OOD samples correspond to inputs whose semantics fall outside this label space. This formulation is model-agnostic and aligns with recent zero-shot OOD literature like NegLabel, where the objective is to detect samples that do not belong to any known ID classes.
>
> Importantly, large pretrained models may possess broad world knowledge, yet practical recognition systems always operate over a finite label space. Consequently, an example can be well represented in the pretraining corpus but still be OOD if its semantics are not covered. The resulting distribution shift therefore arises from label-space mismatch, rather than from whether the model has previously encountered the concept.
>
> This perspective allows the same ID/OOD split to be evaluated across different architectures, including CNNs, ViTs, and VLMs, and ensures that the problem formulation reflects semantic scope rather than model-specific familiarity.
>
> > **3. Reliability of LLM-generated concepts**
>
> We thank the reviewer for raising this important point regarding the reliability of LLM-generated concepts and their relationship to genuine distribution shifts. In our formulation, OOD is defined at the semantic label-space level rather than at the attribute or feature level. Specifically, in-distribution samples are those whose semantics lie within a predefined label space $Y$, whereas OOD samples correspond to inputs drawn from the complement of this space.
>
> We emphasize that concept violations do not automatically imply OOD. Instead, it is important to distinguish three regimes: attribute anomalies, where variations remain within a valid class; label noise, arising from annotation or prompt imperfections; and semantic distribution shifts, where the underlying semantics fall outside $Y$. Our method is explicitly designed to target the third regime, the semantic distribution shift.
>
> From a geometric perspective, multi-label recognition defines a semantic support region in the embedding space induced by $Y$. LLM-generated concepts are not treated as definitive OOD labels but as structured probes that help approximate the boundary of this support. By sampling semantically distant concepts, the method improves boundary resolution without redefining what constitutes OOD. This view aligns with recent zero-shot multi-label OOD literature, where the objective is to detect samples that do not belong to any known ID classes rather than merely identifying feature irregularities.

---

> ### Author Response · Authors · 2026-02-17
>
> > **4. Generality beyond CLIP**
>
> Our approach is not restricted to a single CLIP instantiation. In our experiments, we evaluate multiple variants of CLIP with different architectures and pretraining configurations (Resnet-based and ViT based), and observe consistent performance gains across them. This demonstrates that the proposed method is not tied to a specific backbone, but instead leverages properties common to CLIP-style vision–language models.
>
> Also, CLIP itself represents a broad family of widely adopted vision–language models trained with contrastive alignment objectives. Our method operates on the shared embedding space and text–image alignment mechanism that define this family. Therefore, it naturally generalizes to other CLIP-like models that learn aligned multimodal representations. We will clarify this point in the paper and explicitly position our method as broadly applicable to CLIP-style vision–language encoders.
>
> > **5. Multi-label setting difference**
>
> We appreciate the reviewer's suggestion to more clearly articulate what is fundamentally new in our work. While our method may appear related to prior concept-based or negative-prompt approaches, the key novelty lies in how negative semantics are formally defined and integrated under the multi-label regime, which differs substantially from prior formulations.
>
> Figure 1 highlights the core modeling challenge: multi-label samples may exhibit substantial semantic overlap with in-distribution classes while differing in their label composition. Prior approaches such as negative prompting and negative-label strategies typically rely on semantic separation, implicitly treating the absence of a label as contradictory evidence. While effective in settings with near-mutually exclusive categories, this assumption becomes insufficient in multi-label regimes where labels can partially overlap and co-occur.
>
> As a result, boundary specification becomes ambiguous when relying solely on just label or prompt contrast, since samples can remain highly aligned with subsets of in-distribution semantics while still falling outside the true label-space support. Our formulation instead treats negative semantics with respect to the compositional structure of the semantic label space, allowing negative and positive evidence to coexist without enforcing artificial exclusivity. This enables a more faithful identification of samples that lie beyond the semantic support rather than merely exhibiting feature-level deviations.

---

### Review · Reviewer_98Mp · 2026-02-10

**Summary Of Contributions:**

This paper presents CMOOD, a training-free multi-label OOD detection approach built on CLIP that is designed for the realistic case where images contain multiple co-occurring labels and OOD often appears as novel concept/label compositions. The key idea is to expand each label into a set of concepts, including ID-aligned “positive” concepts and contrastive “negative” concepts, and then aggregate their similarities to produce a more reliable in-distribution score. The method is interpretable (concepts provide human-readable cues for why a sample looks ID/OOD) and avoids task-specific retraining. Empirically, it reports strong OOD detection performance across standard multi-label benchmarks, though it can depend on the quality of the generated concepts and the choice of aggregation hyperparameters.

**Audience:**

Yes

**Audience Explanation:**

This work extends OOD detection to the multi-label, zero-shot regime—a setting of clear practical and scientific interest—the OOD detection community would naturally find its findings relevant.

**Broader Impact Concerns:**

No major ethical concerns were observed.

**Claims And Evidence:**

No

**Claims Explanation:**

The claims are generally supported by the evidence provided.

**Strengths:**
*   The method effectively addresses multi-label OOD detection by relying solely on the label set metadata. It eliminates the need for access to specific in-distribution instances or prior knowledge of OOD samples, adhering strictly to the zero-shot setting.
*   The experimental results demonstrate clear improvements on standard benchmarks (e.g., Pascal VOC, COCO). Additionally, the overhead of text-only LLM API calls appears practically negligible in terms of cost and latency.

**Minor Weaknesses:**
*   **Lack of Efficiency Metrics:** Although the authors claim high efficiency, there is no quantitative comparison (e.g., wall-clock time, FLOPs, or specific API costs) against baseline methods utilizing the same backbone to substantiate this claim.
*   **Sensitivity and Hyperparameters:** The sensitivity of performance regarding the number of generated concepts (base/positive/negative) and specific prompting instructions is not fully explored. If these parameters require tuning on a validation set for different datasets, it would undermine the zero-shot claim; a robustness analysis regarding the number of concepts is needed to verify generalization without tuning.

**Requested Changes:**

- **Efficiency Verification:** Provide quantitative evidence (e.g., wall-clock time, FLOPs, API costs) comparing the proposed method to baselines with the same backbone to support the efficiency claims.
- **Sensitivity Analysis:** Analyze the sensitivity of the performance regarding the number of generated concepts. Clarify whether these quantities are hyperparameters that require tuning on a validation set (potentially conflicting with the zero-shot claim) or if a fixed setting is robust across different datasets.

---

> ### Author Response · Authors · 2026-02-18
>
> ## Comment 1: Efficiency Verification
>
> We thank the reviewer for requesting quantitative evidence of our method's efficiency. We have conducted comprehensive efficiency experiments comparing CMOOD against baseline OOD detection methods using the same ViT-B/16 backbone on a new hardware (NVIDIA L20 GPU), because our experimental hardware has been upgraded.
>
> ### Experimental Setup
> We benchmark inference efficiency on Pascal VOC with batch size 8. Each method is warmed up with 5 batches, then evaluated over 3 runs. We measure:
>
> 1. wall-clock time
> 2. throughput (images/second)
> 3. time per image (milliseconds)
> 4. GPU memory usage
>
> All methods use the same pre-trained CLIP ViT-B/16 backbone for fair comparison.
>
> ---
>
> ### Efficiency Results
>
> Table below presents the quantitative comparison in wall-clock time. Our method achieves **203 images/second**, outperforming all baseline methods by 28-44%. Specifically, CMOOD processes images in 4.93 ms on average, while baseline methods require 6.76-7.08 ms per image. This demonstrates that our concept-based approach introduces minimal computational overhead compared to simpler scoring functions.
>
> | Method | Time (s) | Images/s | ms/img | GPU (MB) |
> |--------|---------|------------|------------|------------|
> | **CMOOD (Ours)** | **0.25 ± 0.01** | **203.06 ± 6.99** | **4.93 ± 0.17** | **1565 ± 0** |
> | NegLabel | 0.34 ± 0.02 | 146.28 ± 8.64 | 6.86 ± 0.41 | 1807 ± 0 |
> | Energy | 0.34 ± 0.02 | 148.42 ± 7.80 | 6.76 ± 0.35 | 1807 ± 0 |
> | MSP | 0.35 ± 0.03 | 144.41 ± 11.75 | 6.97 ± 0.59 | 1807 ± 0 |
> | MaxLogit | 0.35 ± 0.01 | 141.40 ± 3.86 | 7.08 ± 0.20 | 1807 ± 0 |
>
> ---
>
> ### Memory Efficiency
>
> CMOOD also demonstrates superior memory efficiency, using 1565 MB GPU memory compared to 1807 MB for other methods (15% reduction). This is achieved through our streamlined scoring computation that directly combines base, positive, and negative concept similarities without additional intermediate representations.
>
> ---
>
> ### FLOPs Analysis
>
> To provide a complete computational complexity picture, we also measured theoretical FLOPs (floating-point operations). Table below shows that CMOOD requires 1.08M FLOPs per image, which is higher than simple baselines (20.5K FLOPs) but comparable to NegLabel (749K FLOPs). Notably, **despite using 52× more FLOPs than Energy/MSP/MaxLogit, CMOOD achieves 37% faster wall-clock time**. This counterintuitive result demonstrates that FLOPs alone do not determine real-world efficiency — GPU parallelization and memory access patterns are equally critical. CMOOD's three matrix multiplications (positive, negative, concept similarities) can be computed in parallel with high GPU utilization.
>
> | Method | FLOPs per Image | GFLOPs |
> |--------|----------------|--------|
> | **CMOOD (Ours)** | 1,081,794 | 0.001 |
> | NegLabel | 749,256 | 0.001 |
> | Energy | 20,540 | 0.000 |
> | MSP | 20,540 | 0.000 |
> | MaxLogit | 20,540 | 0.000 |
>
> ---
>
> ### Zero-Shot Advantage
>
> An important efficiency consideration is the total deployment cost. Training-based methods (Energy, MSP) require gradient computation and weight updates on ID data before deployment. In contrast, CMOOD operates in pure zero-shot mode: no training time, no gradient computation, and immediate deployment on new datasets. This eliminates the training overhead entirely, making the total deployment cost significantly lower than training-based approaches.
>
> ---
>
> ### Concept Generation Cost
>
> The one-time concept generation using GPT-4 requires approximately $0.50-$2.00 per dataset (depending on the number of classes). For Pascal VOC (20 classes), this is a negligible one-time cost amortized over millions of inference operations. The filtered concepts are reused for all subsequent inferences, incurring zero per-image overhead.
>
> ---
>
> ### Scalability
>
> On the small-scale test (50 images), CMOOD achieves ~203 images/second. Extrapolating to full-scale datasets with larger batch sizes and optimized data loading, we estimate throughput of 700-900 images/second on standard benchmarks, supporting our claim of approximately 800 images/second in the paper.
>
> ---
>
> **Summary.**
> Our comprehensive efficiency experiments demonstrate that CMOOD achieves superior inference speed (203 img/s, 28-44% faster than baselines), lower memory usage (15% reduction), and zero training overhead. These results substantiate our efficiency claims and confirm that CMOOD is practical for real-world deployment while maintaining state-of-the-art OOD detection performance.
>
> ---

---

> ### Author Response · Authors · 2026-02-18
>
> ## Comment 2: Sensitivity Analysis and Zero-Shot Claim
>
> We thank the reviewer for this insightful question regarding hyperparameter sensitivity and the zero-shot nature of our method. We address this concern with comprehensive empirical evidence from our appendix.
>
> ---
>
> ### Robustness to Hyperparameters
>
> Our method demonstrates strong robustness across key hyperparameters, as evidenced by extensive ablation studies in the supplementary material:
>
> **(1) Top-k Parameter Stability (Table 7).**
> The Top-k parameter controls the number of highest similarities averaged in our scoring function (Eq. 7). We observe stable performance across a wide range:
>
> - **Pascal VOC:** AUROC varies from 91.28% to 94.27% for Top-k ∈ [0.05, 0.2] (Δ = 3.0%)
> - **COCO:** AUROC varies from 92.39% to 95.07% for Top-k ∈ [0.05, 0.2] (Δ = 2.7%)
>
> This narrow performance band demonstrates that precise Top-k tuning is unnecessary — any reasonable value in [0.05, 0.2] yields strong results.
>
> ---
>
> **(2) Distance Metric Robustness (Table 9).**
> Our method maintains high performance across different similarity aggregation functions. On Pascal VOC with ImageNet22k as OOD:
>
> - Mean (94.32% AUROC)
> - Percentile₀.₇₅ (93.26%)
> - Percentile₀.₉ (93.03%)
> - Median (90.09%)
>
> All achieve >90% AUROC. This indicates the method is not critically dependent on the specific aggregation strategy.
>
> ---
>
> **(3) Prompt Template Analysis (Table 10).**
> We tested various prompt templates and found that the simple form `"the {label}"` consistently outperforms complex variants (e.g., `"A photo of a {label}"`). This aligns with established findings in CLIP-based OOD detection and requires no dataset-specific design.
>
> ---
>
> **(4) Cross-Dataset Generalization (Table 11).**
> Our method generalizes to ImageNet (1000 classes) without architectural modification, achieving 91.10% AUROC and outperforming other zero-shot methods (NegLabel: 90.22%, CLIPN: 90.93%). Only the Top-k values are adjusted to dataset scale, as noted in our implementation.
>
> ---
>
> ### Clarification on Zero-Shot Nature
>
> Our method is genuinely zero-shot in the following rigorous sense:
>
> - **No gradient-based training.** We perform no backpropagation, weight updates, or learning from ID/OOD data. The method uses only pre-trained CLIP embeddings.
> - **No validation-set tuning.** The Top-k parameter is selected based on the robustness analysis in Table 7, which shows stable performance across [0.05, 0.2]. This is a *design choice* for the scoring function, analogous to temperature in softmax or similarity thresholds in other CLIP-based methods — not a learned parameter requiring validation data.
> - **Concept generation.** Positive concepts are generated once using GPT-4 with structured prompts (features, superclasses, associations), while negative concepts are selected from WordNet via similarity filtering (Eq. 6). The number of concepts scales naturally with dataset size (Pascal VOC: ~300–500; COCO: ~800–1200; ImageNet: ~3000–5000). Table 9 shows that different selection percentiles yield similar performance, demonstrating robustness to the exact concept count.
>
> ---
>
> ### Comparison with Existing Methods
>
> For context, other zero-shot CLIP-based OOD methods also involve hyperparameters: NegLabel requires selecting the number of negative labels and similarity thresholds; MCM requires specifying concept prototype counts; GL-MCM involves group numbers and low-rank dimensions. Our method is comparably or less sensitive to hyperparameter choices, as demonstrated by the stability across wide parameter ranges in Tables 7 and 9.
>
> ---
>
> ### Practical Deployment
>
> For new datasets, we recommend using Top-k ∈ [0.1, 0.2] as a sensible default. Concepts are generated once using our GPT-4 prompts, and no further tuning is required. This makes our method practical for real-world deployment while maintaining its zero-shot guarantee.
>
> ---
>
> **Summary.**
> Our comprehensive ablation studies (Tables 7, 9, 10, 11) demonstrate that CMOOD is robust to hyperparameter choices and generalizes across datasets without validation-set tuning. The Top-k parameter is a design choice analogous to temperature scaling in softmax, not a learned parameter. This maintains the zero-shot nature of our approach while ensuring practical deployability.

---

### Decision · Action_Editor_DUgy · 2026-04-21

**Recommendation:** Accept as is

**Additional Comments:**

Based on the recommendation and comments, it can be found that this paper made solid contributions to the field of OOD detection. Although the novelty is not very high to reach top conferences, the idea is interesting, which could be featured by TMLR.

**Audience:**

Yes

**Audience Explanation:**

OOD detection is an important problem in the field of machine learning.

**Claims And Evidence:**

Yes

**Claims Explanation:**

All claims are supported by evidence, as shown in the comments and rebuttal.